# FROM TRANSFORMER TO TRANSPONDER: INTRODUCING MODULATION TRAINING FOR BIO-INSPIRED META-LEARNING IN LLMS

## ABSTRACT

Transformers are the backbone of state-of-the-art systems across language, vision, and multimodal learning tasks, yet the relevance scale of their functional blocks (self-attention and feed-forward networks) is typically constant across inputs and depth. Motivated by neuro-glia and epigenetic mechanisms—where glial cells and epigenetic processes modulate when and how neurons or genes express their activity—we introduce the *contextual modulator*: a lightweight, input-aware, neuroglia-inspired meta-learner that rescales the outputs of linear sublayers within a block at token- and channel-level granularity. The modulator is implemented via compact parametric functions and adds negligible parameter overhead. Building on this idea, we propose TRANSPONDER, which integrates contextual modulators throughout Transformer blocks to endow functional residual architectures with fine-grained, input-adaptive control. TRANSPONDER provides evident improvement over other scaling or gating methods across LLaMA backbones ranging from 60M to 1B parameters, yielding consistent perplexity reductions with $\sim 1\%$ additional parameters. Analysis reveals depth-, module-, and token-specific scaling patterns, indicating that learned modulators act as input-adaptive regulators of residual information flow. TRANSPONDER provides a simple, general mechanism for hierarchical meta-learning the base components of the Transformer-based models with context-sensitive modulators, providing robust and significant performance improvements without substantial architectural changes.

## 1 INTRODUCTION

Transformer architectures deliver state-of-the-art performance across vision, language, and multimodal tasks (Vaswani et al., 2017; Devlin et al., 2019; Radford et al., 2018; Touvron et al., 2023). In standard designs, information flows through a sequence of linear and nonlinear components—query/key/value projections, attention output projections, and the two feed-forward networks (FFN) linear maps—interleaved with residual connections. After training, the scaling by which these components contribute to the residual stream is effectively static for all the input tokens. Several mechanisms attempt to manipulate residual or path scaling via global reparameterizations ReZero (Bachlechner et al., 2021), DeepNorm/DeepNet (Wang et al., 2024) or depth-aware normalization (Li et al., 2024; Sun et al., 2025). Yet these approaches assign fixed gains in spite of the current token, channel, or depth position.

These designs clash with the core goal of representation learning: the relevance of a component's output is input-dependent. Rare tokens may call for amplifying specific heads; bursty channels may require attenuation; deeper layers may benefit from different mixing strengths than shallow ones. When scaling is static, the model must implicitly encode such selectivity inside the functional maps themselves—mixing representation (what to compute) with control (how much to pass through).

In biological neural systems, neuro–glial circuits implement this meta-regulation process: glial cells modulate when and how much neuronal populations express their activity. Inspired by this separation between representation and regulation, we introduce TRANSPONDER, whose core principle is simple: **pair each Transformer base component with a contextual, input-aware modulator that explicitly controls its contribution at inference**. Specifically, for a component with output

$\mathbf{Y} = \mathbf{F}(\mathbf{X})$, we attach a lightweight modulator $g(\mathbf{X}) \in [0, 1]$ and compute $\tilde{\mathbf{Y}} = \mathbf{F}(\mathbf{X}) \odot g(\mathbf{X})$. This decouples representation from control, turning residual mixing from a static heuristic into a principled, context-sensitive regulation mechanism. We define this process as a **hirechical meta-learning process**. Compared to standard meta-learning in machine learning, which is usually defined over a distribution of tasks and aims to learn how to learn across task shifts (Finn et al., 2017; Hospedales et al., 2021; Thrun & Pratt, 1998), our use of meta-learning in TRANSPONDER is hierarchical rather than across tasks. In our setting, there is a single main task (e.g., language modeling), but the optimization is decomposed into two coupled levels: (1) the base Transformer layers learn task-specific representations (the usual optimization objective), and (2) the TRANSPONDER modulators learn how to help the base layers learn, by contextually modulating their activations and gradients. The modulators are not optimized to solve the prediction task directly; instead, they are optimized for their indirect effect on the performance of the base network, i.e., to improve how the base layers are trained and used. In this sense, TRANSPONDER implements a form of *hierarchical meta-learning*: the lower level learns the task itself, while the higher level (the modulators) learns how to organize, regulate, and adapt the learning dynamics of the base layers, analogous to learning how to learn but applied within a single task rather than across multiple tasks.

We instantiate this principle with TRANSPONDER, designed to be minimally invasive and broadly compatible:

1. **Input-aware modulation.** Modulators calculate the scales that multiply functional-path outputs at use time, yielding context-conditioned scaling factors for each component in Transformer.

2. **Granularity.** For each linear layer, Transponder uses two complementary modulators. The channel-wise modulator provides a fine-grained controller that selectively amplifies or attenuates individual feature channels, while the global scalar modulator provides a coarse, layer-level controller that scales the overall contribution of the transformation.

3. **Low overhead, easy integration.** TRANSPONDER adds at most $\leq 1\%$ additional parameters and integrates into standard Transformer blocks with minimal code changes. We implement each modulator via a fused Triton kernel, keeping inference speed similar to the original model.

Across large-scale language modeling benchmarks, TRANSPONDER delivers consistent gains on LLaMA backbones from 60M to 1B parameters, achieving relative perplexity reductions of 5.8–15.3% over the corresponding baselines. Extensive ablations over modulator placement, granularity, hidden size, and component coverage (Self-Attention and FFN) demonstrate the robustness of the design, and gradient-norm analysis further indicates that TRANSPONDER enables a more stable training process.

## 2 RELATED WORK

### 2.1 RESIDUAL CONNECTIONS AND THE TRANSFORMER ARCHITECTURE

Residual connections (He et al., 2016) enable very deep models by adding a learned transformation to an identity shortcut, improving gradient flow, preserving signal, and allowing layers to refine rather than reconstruct representations. The Transformer (Vaswani et al., 2017) instantiates this principle with a persistent residual stream that carries token-wise state across layers while each block applies a functional path—multi-head self-attention (MHA) followed by a position-wise feed-forward network (FFN). Concretely, each block computes linear projections for queries, keys, and values and an attention output, then applies two FFN linear maps with nonlinearities; the results are mixed back into the residual stream through additive shortcuts. Layer normalization (LayerNorm) is interleaved with these sublayers: the original Post-LN normalizes after each sublayer (Vaswani et al., 2017), whereas Pre-LN normalizes before sublayers, improving optimization stability for deeper stacks (Xiong et al., 2020). This decomposition—identity carryover plus functional updates—yields strong gradient propagation and composability, but it also implicitly fixes the contribution of each subcomponent to the residual stream at inference, motivating methods that explicitly control (or scale) block updates relative to the identity path.

## 2.2 SCALING THE BLOCK: STATIC REPARAMETERIZATIONS AND NORMALIZATION PLACEMENT

A substantial line of work manipulates the magnitude of each block's update relative to the residual stream via static, input-agnostic mechanisms. ReZero (Bachlechner et al., 2021) introduces a zero-initialized, learnable scalar per block that gates the residual branch, effectively starting from an identity mapping and allowing depth to emerge during training. DeepNorm/DeepNet (Wang et al., 2024) analytically rescales residual and sublayer outputs to maintain stability in very deep Transformers, enabling substantially deeper stacks without divergence. LAuReL (Menghani et al., 2024) generalizes residual/functional mixing with learned coefficients (e.g., RW/LR/PA variants), providing a tunable trade-off between identity flow and the functional path. While these approaches improve training dynamics and depth, their coefficients are typically block-level and input-agnostic at inference time, leaving the strength of updates fixed for any given input.

## 2.3 CONTEXT-CONDITIONED GATING METHODS

In convolutional networks, squeeze-and-excitation (SE) (Hu et al., 2019) introduces feature-wise, input-conditioned scaling by pooling global context and learning channel gates, improving representational efficiency and accuracy. The core idea—context-aware modulation of features—has influenced architectures beyond convolution. Within Transformers, SDPA-gated mechanisms (Qiu et al., 2025) attach the gating mechanism to scaled dot-product attention, commonly parameterized at the head or channels. These gates selectively modulate attention behavior and can stabilize training, but they often incur heavy additional parameters and computation, and primarily target the attention pathway.

# 3 TRANSPONDER - CONTEXTUAL MODULATION TRAINING

## 3.1 PRELIMINARY: BIOLOGICAL INSPIRATION

As illustrated in Figure 1, our modulators are conceived as a meta-layer that controls the expression of each component's computation, in analogy with how neuro–glial and epigenetic processes regulate activity in biological systems. Glial cells are now recognized as active regulators of information processing rather than passive support: they modulate synaptic efficacy, neuronal excitability, and local circuit states in a context-dependent manner, providing a flexible "control layer" on top of relatively stable synaptic connectivity (Fields, 2004; Araque et al., 2014; De Pittà et al., 2011). Building on this view, several computational frameworks treat astrocytes as higher-order controllers that shape when and how neuronal learning is expressed (De Pittà et al., 2016; Volman et al., 2012). In parallel, epigenetic mechanisms adjust the accessibility and expression level of genes without changing the underlying DNA sequence, thereby modulating long-term, experience-dependent adaptation of gene expression profiles to environmental or developmental context (Bird, 2007; Sweatt, 2013).

TRANSPONDER adopts this neuro–glia and epigenetic perspective by explicitly decoupling *what* the Transformer backbone computes from *how strongly and when* these computations are expressed. Concretely, we introduce lightweight, input-conditioned modulators that scale the functional information flow inside Transformer blocks. The objective is to turn static functional compositions into context-aware modulation while preserving (i) the inductive bias of the base operators (self-attention and MLP), (ii) training stability, and (iii) negligible parameter and FLOPs overhead. The overall architecture of TRANSPONDER and its integration into self-attention and feed-forward sublayers are summarized in Figure 1.

## 3.2 DEFINITION: TRANSPONDER SCHEME

A *Modulator* $\mathcal{G}_l$ produces a contextual scaling signal that modulates a sublayer at its point of use. For a generic operator $\mathcal{F}_l$ acting on $\mathbf{X}$,

$$\mathbf{Y} \; = \; \mathcal{F}_l(\mathbf{X}) \; \odot \; \mathcal{G}_l(\mathbf{X}), \tag{1}$$

In TRANSPONDER, we instantiate $\mathcal{F}_l$ as all the linear projections inside self-attention and feed-forward sublayers (e.g., $Q, K, V, O$, FFN up/gate/down projections), so each linear layer is paired with its own contextual modulator.

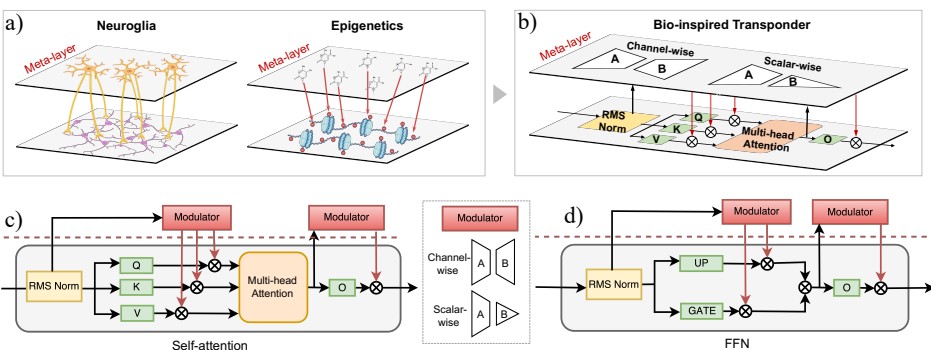

Figure 1: **Bio-inspired meta-layer and the TRANSPONDER mechanism.** (a) Neuro–glial and epigenetic systems can be viewed as meta-layers that regulate when and how underlying neurons or genes express their activity. (b) TRANSPONDER mirrors this design by adding channel-wise and scalar-wise modulators as a meta-layer on top of Transformer components. (c,d) Detailed illustration of where and how the two types of modulators are integrated into self-attention and FFN blocks to modulate the underlying base components.

**Modulator parameterization.** We parameterize the modulator $\mathcal{G}_l$ with a compact bottleneck applied to each input activation $\mathbf{X}$, followed by a bounded nonlinearity:

$$\mathcal{G}_l(\mathbf{X}) \;=\; \tilde{\sigma}_{\alpha_l}\!\Big(\mathbf{B}_l\,\phi\big(\mathbf{A}_l\,\mathbf{X}\big)\Big), \qquad \mathbf{A}_l \in \mathbb{R}^{r \times d_{\text{in}}}, \quad \mathbf{B}_l \in \mathbb{R}^{d_{\text{out}} \times r}, \tag{2}$$

where $d_{\text{in}}$ is the dimension of the input activation, $d_{\text{out}}$ matches the modulation resolution (a single scalar or a channel-wise vector), $\phi$ is a pointwise nonlinearity (e.g., Sigmoid), and $\tilde{\sigma}_\alpha$ is a learnable-curvature sigmoid,

$$\sigma_\alpha(x) \;=\; \frac{1}{1 + \exp(-\alpha x)}, \qquad \alpha > 0, \qquad \tilde{\sigma}_\alpha(x) \;=\; 2\,\sigma_\alpha(x). \tag{3}$$

The base gate $\sigma_\alpha$ maps to $[0, 1]$ and interpolates between soft scaling (small $\alpha$) and near-binary gating (large $\alpha$). We use the calibrated gate $\tilde{\sigma}_\alpha \in [0, 2]$ so that $\tilde{\sigma}_\alpha(0) = 1$, preserving the expected residual magnitude at initialization.

### 3.3 GRANULARITY: WHERE AND AT WHAT RESOLUTION TO MODULATE?

We distinguish *where* the Modulator is applied across the network from *how finely* it is parameterized.

**Placement across layers (inter–layer granularity).** In our main design, modulators are attached to *linear layers* inside each Transformer block. Concretely, after a projection $\mathbf{WX}$ inside a functional path, we apply

$$\underbrace{\mathbf{WX}}_{\text{projection}} \;\mapsto\; \big(\mathbf{WX}\big) \odot \mathcal{G}_l(\mathbf{X}), \tag{4}$$

giving a contextual scale for each projection (e.g., $Q, K, V, O$, FFN up/gate/down) with minimal additional computation. We also experimented with *path-wise* modulation, where a single gate is applied to the entire self-attention or FFN path, but found it to be less stable during training and consistently worse in validation perplexity. We therefore focus on projection-level (per–linear-layer) modulators in the main method and report these ablations in the Section B.1.

**Resolution within a layer (intra–layer granularity).** Let $d_{\text{out}}$ be the dimension of $\mathcal{F}_l$'s output at the modulation site. We instantiate:

- **Layer–wise (scalar) modulation:** $\mathcal{G}_l(\mathbf{X}) \in \mathbb{R}$ via $\mathbf{B}_l \in \mathbb{R}^{1 \times r}$. A single scale uniformly modulates the linear layer; parameter and compute overheads are negligible.
- **Channel–wise modulation:** $\mathcal{G}_l(\mathbf{X}) \in \mathbb{R}^{d_{\text{out}}}$ via $\mathbf{B}_l \in \mathbb{R}^{d_{\text{out}} \times r}$, enabling feature–specific modulation with two low–rank matrices.

### 3.4 FUSED PROJECTION FOR LINEAR-LAYER TRANSPONDER

For each linear layer, the dominant cost is the matrix product $\mathbf{X}\mathbf{W}_\star$. When the summary $u_\star(\mathbf{X})$ is token-wise, we also need $\mathbf{X}\mathbf{A}_\star^\top$. Instead of launching two kernels, we concatenate along the output dimension and perform a single tiled GEMM:

$$\begin{bmatrix} \mathbf{Z}_\star & \mathbf{U}_\star \end{bmatrix} = \mathbf{X} \underbrace{\begin{bmatrix} \mathbf{W}_\star \mid \mathbf{A}_\star^\top \end{bmatrix}}_{\mathbf{C}_\star \in \mathbb{R}^{d_{\text{in}} \times (d_{\text{out}}+r)}}, \qquad \mathbf{Z}_\star \in \mathbb{R}^{B \times T \times d_{\text{out}}}, \ \ \mathbf{U}_\star \in \mathbb{R}^{B \times T \times r}. \tag{5}$$

Here $\mathbf{A}_\star \in \mathbb{R}^{r \times d_{\text{in}}}$ is the rank-$r$ bottleneck, $\mathbf{W}_\star \in \mathbb{R}^{d_{\text{in}} \times d_{\text{out}}}$ is the projection, $B$ is batch size, and $T$ is sequence length. After the fused GEMM, we compute the modulator logits and modulate in-register per tile:

$$\text{logits}_\star = \mathbf{U}_\star \mathbf{B}_\star^\top, \qquad \mathbf{B}_\star \in \mathbb{R}^{d_{\text{out}} \times r}, \qquad \boldsymbol{\gamma}_\star = \tilde{\sigma}_{\alpha_\star}(\text{logits}_\star) \in \begin{cases} \mathbb{R}^{B \times T \times 1}, & \text{scalar gate,} \\ \mathbb{R}^{B \times T \times d_{\text{out}}}, & \text{channel-wise gate.} \end{cases} \tag{6}$$

Finally, we apply broadcasted modulation to the projected activations:

$$\widehat{\mathbf{Z}}_\star = \boldsymbol{\gamma}_\star \odot \mathbf{Z}_\star. \tag{7}$$

This fusion amortizes the summary computation into the main GEMM and keeps the modulator arithmetic on-chip, yielding precise linear-layer control at minimal overhead.

## 4 EXPERIMENTS

### 4.1 SETUP

We evaluate TRANSPONDER on standard language modeling with LLaMA backbones (Touvron et al., 2023) ranging from $\sim$ 60M to $\sim$ 1B parameters, trained on the C4 (Raffel et al., 2020) and OpenWebText (Gokaslan & Cohen, 2019) corpora. We also implement the Transponder in other architectures, like OPT (Zhang et al., 2022) and Transformer (Vaswani et al., 2017), in Appendix F. All models are optimized with Adam and are trained with bfloat16 for all the activations, weights, and optimization states. The hidden dimension of the contextual modulator is set to $r = 8$ in the main comparisons to emphasize gains from contextual modulation rather than capacity. For fairness, we keep all the configs except for the learning-rate schedules identical across baselines and TRANSPONDER. We report the perplexity (the lower the better) as the standard language-modeling metric and measure parameter overhead relative to the corresponding backbone. We initialize weight matrices $A, B$ of the Transponder with Kaiming uniform (He et al., 2015) and zero biases. We set $\alpha=1$ initially for the learnable sigmoid. Detailed hyperparameter settings can be found in Table 9. We explain how to count the extra parameter budget of TRANSPONDER can be found in Appendix H.

### 4.2 BASELINES

In the main text, we primarily compare against the recently proposed gating method SDPA (Qiu et al., 2025), which applies an element-wise sigmoid gate to the scaled dot-product attention output to modulate its contribution and mitigate attention-sink effects. To more broadly compare to this gating mechanism, we further implement an **ALL-Gate** variant that extends the same gating scheme to all linear sublayers, including the MLP projections, thereby providing a stronger and more comprehensive gating baseline. In addition, we compare against widely used residual/normalization designs and several recent residual-scaling or layer-wise contextual control methods. These results are reported in Appendix G due to space constraints. To enable a fair comparison with TRANSPONDER, we also include a LLaMA_widen baseline, obtained by widening the FFN intermediate dimension so that its parameter budget matches that of TRANSPONDER.

### 4.3 MAIN RESULTS FOR LANGUAGE MODELING.

Table 2: Validation perplexity (PPL, lower is better) on C4 and OpenWebText. "Param ↑" denotes the relative parameter increase vs. LLaMA.

| | | 60M | | 130M | | 250M | |
|---|---|---|---|---|---|---|---|
| Training tokens | | 1.2B | | 2.2B | | 3.9B | |
| Dataset | Model | PPL | Param ↑ | PPL | Param ↑ | PPL | Param ↑ |
| OpenWebText | LLaMA (baseline) | 26.56 | - | 19.27 | - | 17.28 | - |
| | LLaMA_widen | 26.59 | 1% | 19.74 | 1% | 17.10 | 1% |
| | SDPA | 22.79 | 4% | 20.44 | 5% | 15.65 | 6% |
| | ALL-Gate | 22.96 | 44% | 17.91 | 65% | 14.97 | 80% |
| | TRANSPONDER | **22.07** | 1% | **17.45** | 1% | **14.93** | 1% |
| C4 | LLaMA (baseline) | 30.31 | - | 26.73 | - | 21.92 | - |
| | LLaMA_widen | 30.05 | 1% | 26.63 | 1% | 21.79 | 1% |
| | SDPA | 29.24 | 4% | 23.92 | 5% | 19.12 | 6% |
| | ALL-Gate | 28.98 | 44% | 22.92 | 65% | 18.82 | 80% |
| | TRANSPONDER | **28.06** | 1% | **21.82** | 1% | **18.72** | 1% |

Table 2 summarizes validation perplexity on Open-WebText and C4 for LLaMA backbones from 60M to 250M parameters, while Table 1 reports results for LLaMA-1B on OpenWebText. Across all model sizes and both datasets, TRANSPONDER consistently achieves the lowest perplexity, despite adding only $\sim 1\%$ additional parameters. In contrast, simply widening the FFN (LLaMA_WIDEN) yields at best marginal gains over the baseline (and sometimes slightly worse), indicating that parameter count alone does not explain the improvements.

Table 1: Perplexity of different methods on LLaMA-1B.

| Model | LLaMA1B |
|---|---|
| LLaMA (baseline) | 14.63 |
| LLaMA_widen | 14.55 |
| SDPA | 13.82 |
| Transponder | **12.51** |

On OpenWebText, TRANSPONDER reduces perplexity from 26.56 to 22.07 at 60M, from 19.27 to 17.45 at 130M, and from 17.28 to 14.93 at 250M, corresponding to relative improvements of roughly 10–20% over the baseline at a fixed parameter budget. Similar trends hold on C4, where TRANSPONDER achieves 28.06, 21.82, and 18.72 perplexity at 60M/130M/250M, outperforming the LLaMA baselines (30.31, 26.73, 21.92) by sizeable margins. These gains persist across different token budgets (1.2B, 2.2B, 3.9B training tokens), suggesting that contextual modulation improves data efficiency as the model and training horizon scale.

**Comparison with gating-based baselines.** Both SDPA and ALL-GATE introduce input-dependent gating, but at a substantially higher parameter cost: 4–6% extra parameters for SDPA and 44–80% for ALL-GATE. While these methods do improve over LLaMA in most settings, TRANSPONDER matches or exceeds their performance with only 1% overhead. For example, on C4 at 250M, TRANSPONDER attains 18.72 perplexity versus 19.12 for SDPA and 18.82 for ALL-GATE, despite using $\sim 6\%$ and $\sim 80\%$ additional parameters, respectively. This indicates that the proposed neuro–glia-inspired modulators provide a more parameter-efficient form of contextual control than simply gating every projection or attention output.

**Extension to 1B parameters.** The LLaMA-1B experiment in Table 1 confirms that these benefits extend to larger scales. TRANSPONDER achieves a perplexity of 12.51, improving over the original LLaMA baseline (14.63) and the parameter-matched LLaMA_WIDEN (14.55), and also outperforming the stronger SDPA baseline (13.82). This corresponds to roughly a 14% reduction in perplexity relative to the standard LLaMA-1B and a notable gain over the best existing gating method. Taken together, these results show that contextual modulators provide consistent, scalable improvements from 60M up to 1B parameters while keeping the architecture and parameter budget nearly unchanged.

Table 4: Validation perplexity (PPL, lower is better) on OpenWebText comparing non-contextual and contextual variants.

|  | 60M | 130M | 250M |
| --- | --- | --- | --- |
| Original LLaMA | 26.56 | 19.27 | 17.28 |
| TRANSPONDER w/o contextual control | 23.40 | 19.29 | 16.29 |
| TRANSPONDER w/o learnable sigmoid | 23.56 | 17.70 | **14.93** |
| TRANSPONDER | **22.50** | **17.45** | **14.93** |

## 4.4 TRANSPONDER STABILIZES OPTIMIZATION

**Stabilizing training via smoother gradients.** In Figure 2 we plot the per-step input gradient norm (averaged over all 7 decoder layers) for LLaMA-60M trained for 10k steps with three variants: the parameter-matched baseline (LLaMA_widen), SDPA gated attention, and TRANSPONDER. All methods exhibit a short warm-up phase at the beginning of training, but the baseline and SDPA models show frequent and large gradient spikes throughout training, indicating strong sensitivity to the learning rate and a higher risk of instability or divergence. In contrast, TRANSPONDER maintains low, smooth gradient norms over the entire training trajectory. This directly supports the view that contextual modulators act as stabilizers of the optimization dynamics, allowing the model to safely operate under more aggressive learning-rate regimes.

**Learning-rate robustness.** To quantify this effect, we conduct a learning-rate sweep on LLaMA-60M and LLaMA-130M using three learning rates ($1 \times 10^{-2}$, $3 \times 10^{-3}$, $1 \times 10^{-3}$) on both OpenWeb-Text (OWT) and C4. For each setting, we compare LLaMA_widen (a parameter-matched baseline obtained by widening the FFN intermediate dimension), SDPA, and TRANSPONDER. Results are summarized in Table 10.

At high learning rates ($1 \times 10^{-2}$ and $3 \times 10^{-3}$), LLaMA_widen and SDPA often exhibit signs of training failure, whereas TRANSPONDER consistently achieves low PPLs. For LLaMA-60M on both OWT and C4, TRANSPONDER achieves its best performance at the largest learning rate ($1 \times 10^{-2}$), while the baselines degrade catastrophically under the same setting. For LLaMA-130M, TRANSPONDER matches or surpasses the best baseline perplexity across all datasets and learning rates, with its strongest improvements at $3 \times 10^{-3}$. Overall, these results show that TRANSPONDER is substantially more robust to large learning rates and can even *benefit* from more aggressive optimization, unlike the baseline and SDPA, which are markedly more fragile.

**Stability for Post-LN.** Beyond the Pre-LN setting, we evaluate TRANSPONDER under the more delicate Post-LN regime. As shown in Table 3, vanilla LLaMA with Post-LN is unstable at scale: PPL explodes at 250M (1409.79), and even at 130M performance degrades to 26.95 in comparison to the Pre-LN LLaMA. In contrast, TRANSPONDER stabilizes training and improves accuracy in both cases,

Table 3: Validation perplexity of LLaMA 130M and 250M on C4 with Post-Layer Norm.

|  | 130M | 250M |
| --- | --- | --- |
| LLaMA w. Post-LN | 26.95 | 1409.79 |
| TRANSPONDER w. Post-LN | **25.71** | **20.28** |

yielding a **4.6%** PPL reduction at 130M (26.95→25.71) and preventing divergence at 250M, where it achieves **20.28** PPL instead of catastrophic failure. While Post-LN remains slightly behind strong Pre-LN baselines at a similar scale, these results indicate that TRANSPONDER substantially enlarges the viable training regime for Post-LN models with only $\sim 1\%$ overhead, mitigating the well-known optimization fragility of Post-LN Transformers.

## 4.5 ABLATIONS AND SENSITIVITY TESTS

**Contextual vs. Non-Contextual.** To assess the contribution of contextual modulation, we compare TRANSPONDER with and without contextual signals. In the non-contextual variant, for each layer $l$, we replace the contextual pathway with learnable parameters—a layer-wise scalar $\beta_1^{(l)}$ and

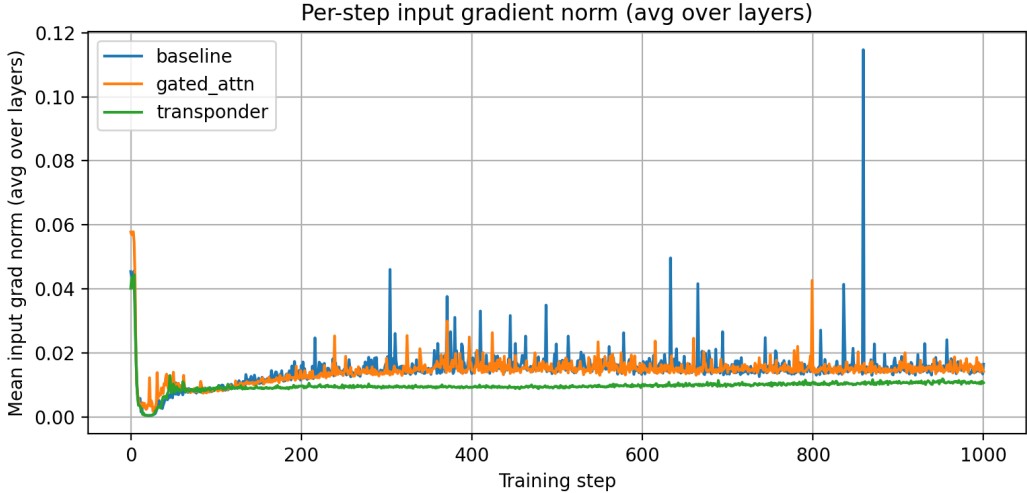

Figure 2: **Per-step input gradient norm on LLaMA-60M.** Mean input gradient norm (averaged over all 7 decoder layers) during training for LLaMA-60M under three variants: the baseline model, the gated_attn (SDPA), and TRANSPONDER.

a channel-wise vector $\boldsymbol{\beta}_2^{(l)}$—and set

$$\mathcal{G}_l \;=\; \tilde{\sigma}_{\alpha_l}\left(\beta_1^{(l)} \cdot \boldsymbol{\beta}_2^{(l)}\right),$$

where $\beta_1^{(l)}$ applies uniform scaling and $\boldsymbol{\beta}_2^{(l)}$ provides per-channel modulation.

As shown in Table 4, the non-contextual variant already improves over the LLaMA baseline at certain scales, confirming the utility of the proposed modulator design. However, adding contextual control consistently delivers much larger gains across all model sizes, establishing it as the key driver of performance. This result directly supports our initial motivation: each component benefits from input-aware, contextual control rather than relying solely on static modulation.

**Sigmoid and Learnable Sigmoid.** A key innovation of TRANSPONDER is the introduction of a learnable sigmoid. This learnable non-linear modulation mechanism allows each modulator to dynamically adjust its sensitivity to the input, either amplifying or attenuating modulation strength as needed. As shown in Table 4, incorporating the learnable sigmoid consistently improves (except for 250M that is comparable) perplexity across model sizes, validating our design intuition that adaptive nonlinearity provides a crucial layer of flexibility for effective modulation.

**Hidden dimensions.** We further examine the effect of the hidden dimension $r$ in the contextual modulator. Table 6 shows that even with an extremely compact setting ($r = 2$), TRANSPONDER already delivers large gains over the LLaMA baseline. Scaling up the hidden size from 8 to 32 provides marginal yet consistent improvements. These findings suggest that the modulator does not require a large intermediate capacity to capture input-dependent scaling, underscoring the efficiency of our design: lightweight modulators suffice to model contextual dependencies while adding negligible parameter overhead.

**Contribution of functional components.** We next investigate which linear module combination benefits most from modulation by selectively applying contextual modulators to individual sub-components (Table 5). Applying modulators to either the self-attention or MLP block alone yields clear improvements, while restricting modulation to partial components (e.g., only_first, only_last, only_qk) provides only limited gains.

We further evaluate the effect of removing the up- and gate-projections. In the original LLaMA design, the gate projection serves as a channel-wise modulator to the up-projection, though implemented as a full-rank matrix. Interestingly, excluding these projections leads to strong performance,

Table 5: Ablation study on modulators on different functional modules for the TRANSPONDER on C4 dataset. The reported metric is Perplexity. The best performance is highlighted in bold.

| Modules | LLaMA60M | LLaMA130M |
|---|---|---|
| Baseline | 30.31 | 26.73 |
| self-attention | 29.04 | 23.21 |
| mlp | 29.43 | 23.36 |
| only_last | 29.88 | 23.97 |
| only_first | 29.91 | 24.11 |
| only_qk | 30.01 | 24.03 |
| w/o up and gate | 29.12 | 22.93 |
| all | **28.55** | **22.09** |

Table 6: Ablation study on the hidden dimension $r$ of both the channel-wise and scalar-wise modulators for the TRANSPONDER on C4 dataset. The reported metric is Perplexity. The best performance is highlighted in bold.

| Hidden Size | LLaMA60M | LLaMA130M |
|---|---|---|
| Baseline | 30.31 | 26.73 |
| 2 | 28.91 | 23.01 |
| 4 | 28.65 | 22.64 |
| 8 | 28.55 | 22.09 |
| 16 | **28.41** | 22.12 |
| 32 | 28.73 | **22.07** |

Table 7: Per-token FLOPs for LLaMA-60M (seq. len. 256).

| Model | GFLOPs/token | |
|---|---|---|
| | Inference | Training |
| LLaMA baseline | 0.082 | 0.245 |
| SDPA | 0.088 | 0.264 |
| Transponder | 0.084 | 0.252 |

Table 8: Inference throughput (tokens/s; higher is better) on LLaMA-60M (seq. len. 256) on a single A100 80G.

| Model | BS=32 | BS=64 | BS=128 |
|---|---|---|---|
| LLaMA baseline | 707.57 | 721.81 | 748.74 |
| SDPA | 672.98 | 688.83 | 707.66 |
| Transponder | 654.53 | 678.96 | 689.76 |

second only to the full "all" configuration. This suggests that the gate-projections already play a key role in channel-wise contextual modulation. We conduct more quantitative research on the effect of gate_proj and our modulators in Appendix I.

Overall, the best results are consistently achieved when all components are modulated jointly ("all"), yielding the lowest PPL on both LLaMA-60M and LLaMA-130M. This confirms that full-path functional modulation is necessary to provide comprehensive control over residual transformations, enabling robust and consistent improvements across scales.

### 4.6 COMPUTATIONAL OVERHEAD AND INFERENCE LATENCY

To quantify the computational cost of TRANSPONDER, we report both per-token FLOPs and inference latency, and compare them to the original LLaMA-60M and the GATED ATTN baseline. All measurements are taken on LLaMA-60M; inference latency is evaluated on a single NVIDIA A100 80G GPU with sequence length 256.

**Per-token FLOPs and latency.** As shown in Tables 7 and 8, TRANSPONDER adds only a small computational and latency overhead. Relative to the original model, forward FLOPs increase from 0.082 to 0.084 GFLOPs/token and training FLOPs from 0.245 to 0.252 GFLOPs/token, corresponding to only about a 2–3% overhead, which is smaller than that of GATED ATTN. Inference throughput decreases by roughly 7–9% depending on batch size, but remains comparable to or better than the GATED ATTN baseline. Overall, this overhead is modest relative to the performance gains achieved in perplexity and training stability.

## 5 CONCLUSION

In this article, we introduced TRANSPONDER, a bio-inspired hierarchical meta-learning scheme for Transformer-based language models. By adding a lightweight meta-layer of scalar- and channel-wise modulators on top of standard linear sublayers, TRANSPONDER explicitly decouples *what* each component computes from *how strongly and when* its output is expressed. This design is minimally invasive—requiring only about 1% additional parameters. Empirically, TRANSPONDER consistently improves language modeling perplexity across LLaMA backbones from 60M to 250M parameters on OpenWebText and C4, and scales favorably up to LLaMA-1B. Under a matched parameter bud-

get, it clearly outperforms the widened LLaMA baseline and remains competitive with or superior to much heavier gating method such as SDPA and ALL-GATE, which require 4–80% extra parameters. Gradient-norm analyses and learning-rate sweeps further show that TRANSPONDER stabilizes optimization: it yields smoother gradients, is substantially more robust to large learning rates, and can even benefit from more aggressive optimization regimes where baselines cannot. The limitation of this work can be found in Appendix A.

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

Table 9: **Hyperparameters of LLaMA-60M, LLaMA-130M, and LLaMA-250M on OpenWeb-Text and C4.**

| Hyper-parameter | LLaMA-60M | LLaMA-130M | LLaMA-250M |
|---|---|---|---|
| Embedding Dimension | 512 | 768 | 1024 |
| Feed-forward Dimension | 1376 | 2048 | 2560 |
| Global Batch Size | 512 | 512 | 512 |
| Sequence Length | 256 | 256 | 256 |
| Training Steps | 10000 | 20000 | 40000 |
| Learning Rate | 1e-2 | 1e-2 (3e-3 for OpenWebText) | 3e-3 |
| Warmup Steps | 1000 | 2000 | 4000 |
| Learning Rate Decay Method | noam | noam | noam |
| Optimizer | Adam | Adam | Adam |
| Number of Layers | 8 | 12 | 24 |
| Number of Heads | 8 | 12 | 16 |

## A  LIMITATION

Our study is compute-limited: we did not scale beyond 1B parameters or train for insufficient tokens because of the limits of the computational resources. Also, as shown in Section 4.6, there is still an around 7-9% of downgrade of thoughput. Future work should evaluate TRANSPONDER at larger scales and with substantially longer training runs (more steps and tokens), and integrate a more engineering impelmentation to speed up.

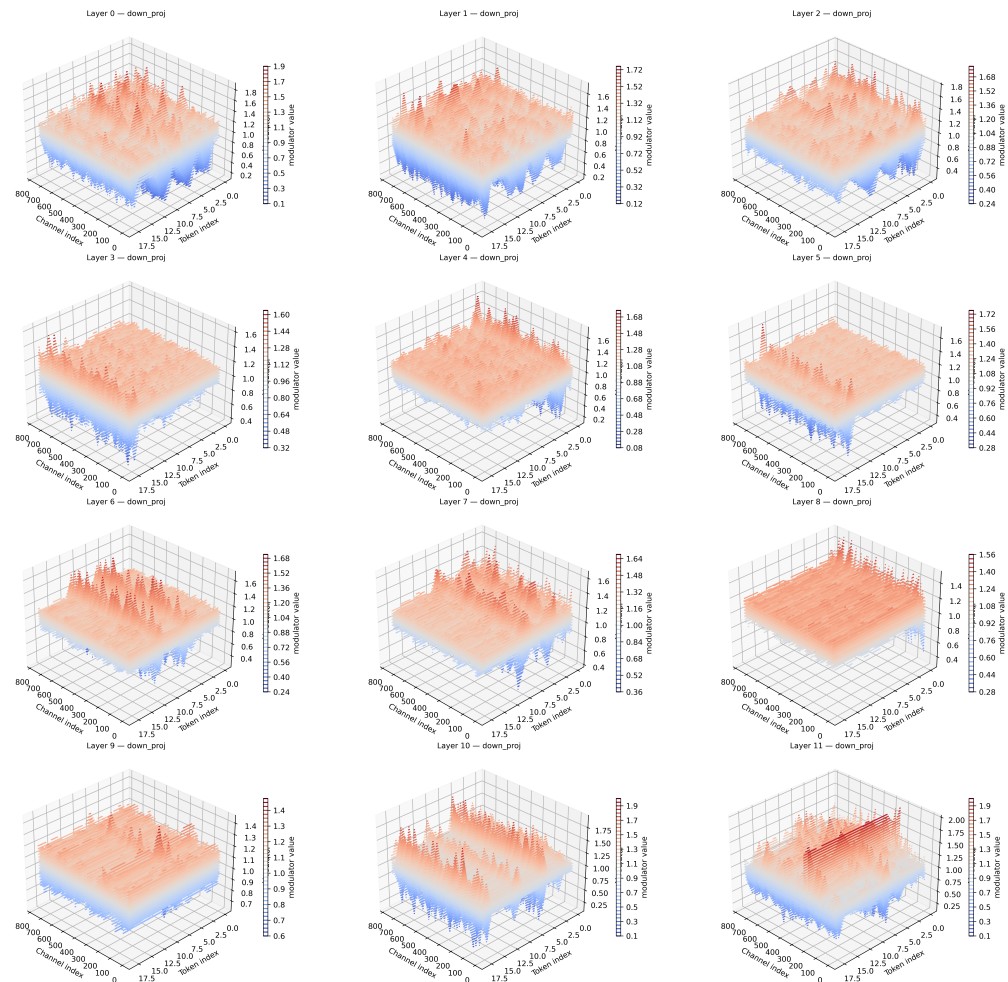

Figure 3: Channel-wise modulator values on down_proj across layers and tokens in a LLaMA-130M model trained with TRANSPONDER. The X and Y are the channel index and the token index, while the Z-axis are the channel-wise modulator values.

## B EXTRA RELATED WORKS

Hyper-Connections (Zhu et al., 2025) generalize residual connections by maintaining a stack of hidden states and learning static or input-dependent matrices ("dynamic hyper-connections") that mix features both across depth and width; in the dynamic variant, the coefficients $(B(H), A_m(H), A_r(H))$ are predicted from the current hyper-state and applied to all layers' hidden vectors. DeepCrossAttention (Heddes et al., 2025) builds on Generalized Residual Networks (GRN-v1/v2/v3) to form input-dependent linear combinations of all previous layer outputs and uses these stack-based GRNs to generate Q/K/V for each block, thereby implementing depth-wise cross-attention and dynamic layer re-composition. Gated Attention (Qiu et al., 2025) systematically studies gating positions inside softmax attention and finds that applying a head-specific sigmoid gate to the SDPA output (or values) improves perplexity and stability for 1.7B–15B LLMs, primarily by adding non-linearity and query-dependent sparsity within the attention block.

### B.1 MODULATOR VARIANTS VALIDATION

We next investigate where modulators should be placed, and at what granularity, to maximize performance. All variants are trained on OpenWebText with identical hyperparameters, including a peak

Table 10: Perplexity (lower is better) under different learning rates for LLaMA-60M and LLaMA-130M on OpenWebText (OWT) and C4. LLaMA_widen is a parameter-matched baseline obtained by widening the FFN intermediate dimension. The best performance under each case is highlighted in bold.

| Model Size | Dataset | Model | 1e-2 | 3e-3 | 1e-3 |
|---|---|---|---|---|---|
| 60M | OWT | LLaMA_widen | 1007.00 | 600.88 | 26.59 |
| | | SDPA | 524.93 | 22.79 | 25.18 |
| | | TRANSPONDER | **22.07** | 22.65 | 25.85 |
| 60M | C4 | LLaMA_widen | 986.52 | 540.23 | 33.05 |
| | | SDPA | 693.65 | 29.24 | 31.84 |
| | | TRANSPONDER | **28.06** | 28.53 | 32.24 |
| 130M | OWT | LLaMA_widen | 683.08 | 632.76 | 19.74 |
| | | SDPA | 610.09 | 20.44 | 18.60 |
| | | TRANSPONDER | 18.49 | **17.49** | 18.26 |
| 130M | C4 | LLaMA_widen | 775.34 | 669.47 | 26.63 |
| | | SDPA | 724.76 | 302.44 | 23.92 |
| | | TRANSPONDER | **21.82** | 22.13 | 23.99 |

Table 11: Validation perplexity (PPL, lower is better) of different variants of contextual modulators on OpenWebText. "Param ↑" denotes the relative parameter increase vs. LLaMA baseline. The best performance is highlighted in bold.

| Model | 60M | | 130M | | 250M | |
|---|---|---|---|---|---|---|
| | PPL | Param ↑ | PPL | Param ↑ | PPL | Param ↑ |
| path-scalar | 62.91 | <1% | 18.29 | <1% | 1088 | <1% |
| path-channel | 23.25 | <1% | 18.91 | <1% | 16.14 | <1% |
| layer-scalar | 23.15 | <1% | 17.76 | <1% | 15.21 | <1% |
| layer-channel | 23.08 | 1% | 17.56 | 1% | 15.17 | 1% |
| layer-channel-scalar (TRANSPONDER) | **22.50** | 1% | **17.45** | 1% | **14.93** | 1% |

learning rate of $3 \times 10^{-3}$. We instantiate five contextual modulator variants that differ in placement (path-wise vs. layer-wise) and resolution (scalar vs. channel-wise):

- **path-scalar**: a single scalar modulator applied to the entire functional path (attention output or MLP output) per block.

- **path-channel**: a channel-wise modulator applied to all output channels of the functional path per block.

- **layer-scalar**: a single scalar modulator per linear projection.

- **layer-channel**: a channel-wise modulator per linear projection.

- **layer-channel-scalar (TRANSPONDER)**: a channel-wise modulator per linear projection combined with an additional scalar output modulator per projection.

Table 11 reveals two consistent trends across all model scales. (i) *Path-wise* modulation is not effective and can even destabilize optimization: the path-scalar variant exhibits abnormally high PPL (e.g., 1088 at 250M), indicating training failure under this learning-rate regime, whereas path-channel offers only limited gains. In contrast, *layer-wise* modulation (layer-scalar and layer-channel) substantially improves both training stability and final perplexity. (ii) Once layer-wise modulators are introduced, combining scalar and channel-wise components further improves performance: the scalar term provides coarse, global control of each projection, while the channel-wise term enables fine-grained adjustments. Using both together (layer–channel–scalar) yields an additional, robust gain under a comparable parameter budget with only $\sim 1\%$ extra parameters.

Given its consistently best validation perplexity under matched computational and parameter budgets, we adopt **layer–channel–scalar** as the default TRANSPONDER configuration for all subsequent comparisons against baseline and the other methods.

## C  DIFFERENCE OF TRANSPONDER AND THE GATING METHODS

Similarly, recent gating-based methods (Touvron et al., 2023; Qiu et al., 2025) also learn input-dependent coefficients that modulate the forward computation, but they differ from TRANSPONDER in several fundamental aspects. In this section, we summarize these differences and clarify why we view Transponder as a meta-learner rather than just another instance of input-aware gating.

**Where the modulating acts.**  Hyper-Connections (Zhu et al., 2025) and DeepCrossAttention (Heddes et al., 2025) operate over stacks of layer outputs and learn how to mix representations across depths (and widths) before feeding them into attention/FFN; Gated Attention (Qiu et al., 2025) modifies the internal attention computation (SDPA/value) but does not change how attention/FFN outputs are injected into the residual stream. By contrast, Transponder keeps the backbone attention and FFN unchanged and instead introduces *residual modulators* that directly control the mixing between each block's output and the main residual stream at layer-wise and channel-wise granularity. This decouples "what is computed" (base Transformer sublayers) from "how strongly it is expressed" (modulators), which is not explicitly addressed in the above works.

**Local vs. cross-layer control and overhead.**  Hyper-Connections (Zhu et al., 2025) and DeepCrossAttention (Heddes et al., 2025) explicitly maintain and re-weight a multi-layer stack, whose mixing patterns span many layers. Transponder uses a local, low-rank contextual network to modulate from the current hidden state only, without storing or re-accessing the full depth history. Consequently, our modulators add 1% parameters while still yielding large PPL improvements in the LLaMA model family; they are closer in spirit to a per-block "controller" for the residual path than to cross-layer aggregation mechanisms. All these methods have small overhead, but the structural role of the added parameters is different.

**Hierarchical Meta-learning and neuro-glia perspective.**  Conceptually, our modulators are designed as a bio-inspired hirechical meta-learning mechanism: a channel-wise modulator and a global scalar modulator modulate the expression of each block's computation, analogous to how glial cells or epigenetic mechanisms regulate when and how underlying neurons/genes express their activity. In earlier drafts we planned to reserve this bio-inspired meta-learning interpretation for a follow-up journal article, where we can fully develop the theory. In light of the reviewer's comment, we will make this conceptual lens explicit in the revised version, clarifying that Transponder can be viewed as a lightweight, neuro-glia-inspired meta-learner on top of a Transformer backbone, rather than merely another form of input-aware gating method.

## D  CONTEXTUAL MODULATOR VALUES ANALYSIS

The results in Section 4 suggest that, beyond static architectural scaling (e.g., fixed layer-normalization schemes), context-/input-sensitive modulation is a powerful way to improve optimization and expressivity in deep language models. TRANSPONDER learns, for each token and sublayer (self-attention and FFN), both a layer scalar and channel-wise modulation. The question is whether these modulators truly encode context rather than acting as static rescalers.

Figure 4 (layer-scalar, per module) shows clear depth- and module-specific structure with token-dependent variation. The traces for different tokens diverge at many layers, indicating context sensitivity. We observe systematic trends across modules: `o_proj` and `down_proj` steadily strengthen with depth, consistent with greater late-layer amplification; `v_proj` exhibits a pronounced surge in upper layers; `up_proj` follows a U-shaped pattern (early attenuation, late amplification); `gate_proj` remains in a tighter band with a mid-depth peak; and `q_proj`/`k_proj` show early suppression followed by recovery.

Figure 3 further confirms the effectiveness of the channel-wise modulation. Early layers are relatively flat, but mid–late layers develop structured ridges over both tokens and channels, with in-

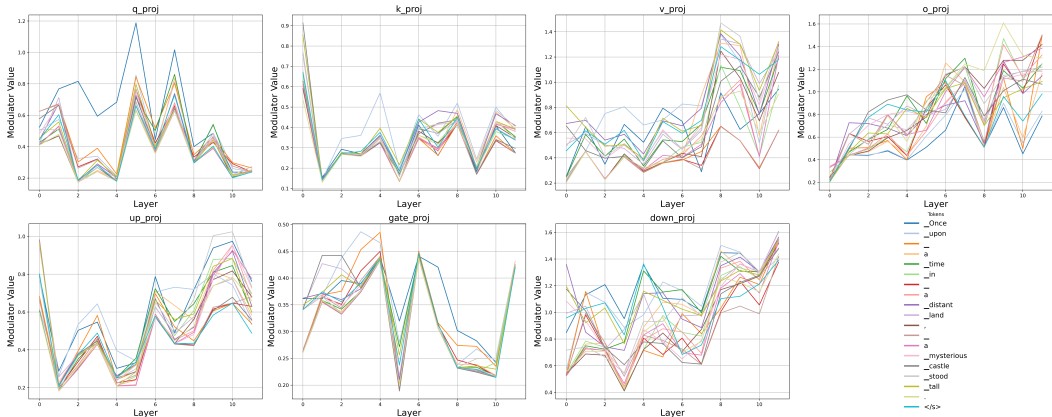

Figure 4: Token-wise modulator values across layers and modules in a LLaMA-130M model trained with TRANSPONDER. We report the values of the scalar modulator as the representative. Each subplot corresponds to a specific linear projection of the decoder layer. The plotted curves represent different tokens from the input sequence.

creasing contrast across depth. In particular, we observe a similar trend at some specific layers. For instance, at the 5th layer, one token exhibits very large modulator values across all channels, whereas in the 11th layer, one channel displays very large modulation values across the tokens.

Taken together, the two views indicate that TRANSPONDER learns structured, context-aware gains that evolve with depth and specialize by module and channel—precisely the behavior we would expect if the modulators were encoding useful contextual signals rather than acting as fixed, global scalars.

## E    EXTENDED TRAINING OF LLAMA-60M

In Table 2, LLaMA-60M is trained for only 10K steps (1.2B tokens), which partly explains its relatively low performance. To further verify the effectiveness of TRANSPONDER, we extend the training to 60K steps (7.78B tokens). Under this stronger training regime, the baseline achieves a perplexity of 22.15 on OpenWebText, while TRANSPONDER achieves 19.56.

Thus, both models can reach perplexity below 22.50 on OWT when trained longer, yet Transponder consistently outperforms the baseline even under extended training.

## F    EXPERIMENTS ON OTHER TRANSFORMER VARIANTS

To verify that Transponder is not restricted to LLaMA-style architectures, we additionally evaluate it on two other Transformer families: (i) a standard encoder–decoder Transformer-Base on machine translation benchmarks, and (ii) OPT-125M (Zhang et al., 2022), a decoder-only model architecturally distinct from LLaMA, on language modeling.

**Transformer-Base on machine translation.**    We evaluate Transponder on three machine translation benchmarks (Multi30K, IWSLT, WMT) using a Transformer-Base encoder–decoder architecture. All results are averaged over three random seeds and we report the mean and standard error. As shown in Table 12, Transponder consistently improves BLEU across all datasets.

**OPT-125M on language modeling.**    We further test Transponder on OPT-125M (Zhang et al., 2022), a decoder-only Transformer that is architecturally distinct from LLaMA. Table 13 reports perplexity (lower is better). Transponder attains the lowest perplexity among all variants.

Table 12: **BLEU scores on machine translation benchmarks using Transformer-Base.** Results are reported as mean $\pm$ standard error over three runs. Transponder consistently outperforms the vanilla Transformer-Base across all datasets.

| Model | Multi30K (BLEU) | IWSLT (BLEU) | WMT (BLEU) |
|---|---|---|---|
| Transformer-Base | $31.38 \pm 0.38$ | $24.48 \pm 0.30$ | 25.22 |
| Transponder | $\textbf{33.03} \pm \textbf{0.29}$ | $\textbf{25.45} \pm \textbf{0.28}$ | **26.08** |

Table 13: **Language modeling perplexity on OPT-125M.** Transponder consistently improves over both the widened baseline and the gated attention variant.

| Model | PPL $\downarrow$ |
|---|---|
| Widen-Original | 25.87 |
| Gated-Attn | 24.97 |
| Transponder | **23.58** |

Overall, these results demonstrate that Transponder generalizes beyond LLaMA-style decoders and remains effective across both encoder–decoder and decoder-only Transformer architectures, as well as across diverse tasks including machine translation and language modeling.

## G COMPARISON WITH THE RESIDUAL SCALING METHODS

We show the comparison with the residual scaling methods in Table 14. DeepNet (Wang et al., 2024), LAuReL-LR (Menghani et al., 2024), LAuReL-PA, and LayerNorm Scaling (Sun et al., 2025) generally provide modest improvements over the LLaMA baseline under similar parameter budgets, but they either fall short of TRANSPONDER or suffer from training instabilities (e.g., very large PPL for LAuReL-PA at 250M on OpenWebText and at 130M on C4). In contrast, TRANSPONDER delivers consistent improvements across all model sizes and datasets with a stable training behavior and only 1% extra parameters.

Under comparable budgets, TRANSPONDER achieves state-of-the-art PPL among all tested variants, with minimal overhead, strong scaling behavior, and improved robustness.

## H 1% EXTRA PARAMETER CALCULATION

A brief summary of the additional parameters introduced by the layer–channel–scalar modulators across all LLaMA scales is provided here.

For a layer with hidden size $d$ and feed-forward dimension $d_{\text{ff}}$, the extra parameters introduced by the modulators are

$$N_{\text{mod}}(d, d_{\text{ff}}) = 4\big(d \times 8 \times 2 + 8 \times 1\big) + 2\big(d \times 8 + d_{\text{ff}} \times 8 + 8 \times 1\big) + \big(d \times 8 + d_{\text{ff}} \times 8 + 8 \times 1\big),$$

which simplifies to

$$N_{\text{mod}}(d, d_{\text{ff}}) = 88d + 24d_{\text{ff}} + 56.$$

Using this expression for each architecture:

- **LLaMA-60M** ($d = 512$, $d_{\text{ff}} = 1376$, $L = 8$):

$$N_{\text{mod}}^{(\text{per layer})} = (512 \times 8 \times 2 + 8) \times 4 + (512 \times 8 + 1376 \times 8 + 8) \times 2 + (512 \times 8 + 1376 \times 8 + 8) = 78{,}136.$$

$$N_{\text{mod}}^{(\text{total})} = 78{,}136 \times 8 = 625{,}088,$$

giving an overhead of

$$\frac{625{,}088}{60{,}000{,}000} \approx 1.04\% \approx 1\%.$$

Table 14: Validation perplexity (PPL, lower is better) on C4 and OpenWebText. "Param ↑" denotes the relative parameter increase vs. LLaMA.

| | | 60M | | 130M | | 250M | |
|---|---|---|---|---|---|---|---|
| Training tokens | | 1.2B | | 2.2B | | 3.9B | |
| Dataset | Model | PPL | Param ↑ | PPL | Param ↑ | PPL | Param ↑ |
| | LLaMA (baseline) | 26.56 | - | 19.27 | - | 17.28 | - |
| | DeepNet | 23.78 | <1% | 18.74 | <1% | 16.53 | <1% |
| | LAuReL-LR | 23.81 | <1% | 18.19 | <1% | 16.75 | <1% |
| OpenWebText | LAuReL-PA | 23.37 | <1% | 18.22 | <1% | 257 | <1% |
| | LayerNorm Scaling | 23.31 | 0% | 18.28 | 0% | 16.16 | 0% |
| | TRANSPONDER | **22.07** | 1% | **17.45** | 1% | **14.93** | 1% |
| | LLaMA (baseline) | 30.31 | - | 26.73 | - | 21.92 | - |
| | DeepNet | 30.18 | <1% | 143.65 | <1% | 21.72 | <1% |
| | LAuReL-LR | 30.05 | <1% | 106.94 | <1% | 39.14 | <1% |
| C4 | LAuReL-PA | 29.50 | <1% | 1355 | <1% | 20.88 | <1% |
| | LayerNorm Scaling | 29.77 | 0% | 25.76 | 0% | 20.35 | 0% |
| | TRANSPONDER | **28.06** | 1% | **21.82** | 1% | **18.72** | 1% |

- **LLaMA-130M** ($d = 768$, $d_{\text{ff}} = 2048$, $L = 12$):

$$N_{\text{mod}}^{\text{(per layer)}} = (768{\times}8{\times}2{+}8){\times}4+(768{\times}8{+}2048{\times}8{+}8){\times}2+(768{\times}8{+}2048{\times}8{+}8) = 116{,}792.$$

$$N_{\text{mod}}^{\text{(total)}} = 116{,}792 \times 12 = 1{,}401{,}504,$$

leading to an overhead of

$$\frac{1{,}401{,}504}{130{,}000{,}000} \approx 1.08\% \approx 1\%.$$

- **LLaMA-250M** ($d = 768$, $d_{\text{ff}} = 2560$, $L = 24$):

$$N_{\text{mod}}^{\text{(per layer)}} = (768{\times}8{\times}2{+}8){\times}4+(768{\times}8{+}2560{\times}8{+}8){\times}2+(768{\times}8{+}2560{\times}8{+}8) = 129{,}080.$$

$$N_{\text{mod}}^{\text{(total)}} = 129{,}080 \times 24 = 3{,}097{,}920,$$

corresponding to an overhead of

$$\frac{3{,}097{,}920}{250{,}000{,}000} \approx 1.24\% \approx 1\%.$$

- **LLaMA-1B** ($d = 2048$, $d_{\text{ff}} = 5461$, $L = 24$):

$$N_{\text{mod}}^{\text{(per layer)}} = (2048{\times}8{\times}2{+}8){\times}4+(2048{\times}8{+}5461{\times}8{+}8){\times}2+(2048{\times}8{+}5461{\times}8{+}8) = 311{,}344.$$

$$N_{\text{mod}}^{\text{(total)}} = 311{,}344 \times 24 = 7{,}472{,}256,$$

yielding an overhead of

$$\frac{7{,}472{,}256}{1{,}000{,}000{,}000} \approx 0.75\% < 1\%.$$

Across all model scales, the additional parameters introduced by TRANSPONDER remain on the order of $\sim 1\%$ of the base model size, and even decrease slightly for the larger LLaMA-1B model.

## I   ABLATION ON LLAMA GATE PROJECTIONS AND TRANSPONDER MODULATORS

To better understand the interaction between LLaMA's native gate_proj and our TRANSPONDER modulators, we conduct a systematic ablation over four configurations: (i) neither gate nor modulators, (ii) gate only (LLaMA baseline), (iii) modulators only, and (iv) both gate and modulators

Table 15: Validation perplexity (PPL, lower is better) for combinations of LLaMA's native `gate_proj` and TRANSPONDER modulators on LLaMA-60M and LLaMA-130M.

| Setting | LLaMA-60M | LLaMA-130M |
|---|---|---|
| w/o gate and modulators | 32.61 | 27.95 |
| w only gate | 30.31 | 26.73 |
| w only modulators | 30.14 | 26.48 |
| w both | **29.91** | **26.26** |

enabled. We report validation perplexity (PPL; lower is better) on LLaMA-60M and LLaMA-130M in Table 15.

Three observations emerge from this study. First, enabling both mechanisms yields the best perplexity at both scales, suggesting that the native gate projections and our modulators are complementary rather than redundant. Second, the *modulator-only* configuration consistently outperforms the *gate-only* configuration, indicating that our low-rank channel-wise modulator combined with a layer-wise scalar modulator provides additional representational and optimization benefits beyond the built-in gating. Third, removing both components leads to the worst performance, confirming that some form of multiplicative control is beneficial and that TRANSPONDER substantially strengthens this effect.

## J    CLAIM OF THE LLM USAGE

We used LLM-based tools to improve the language and flow; the principles, core logic, and innovations are entirely the authors'.

## K    REPRODUCTION STATEMENT

All experiments were conducted using the Hugging Face pretraining framework with data parallelism over $8\times$ NVIDIA A100 (40 GB) GPUs. For DeepNorm and LayerNorm scaling, we report the results from (Sun et al., 2025), as we adopt identical hyperparameter settings. To ensure reproducibility, we include our code and step-by-step instructions in the supplementary materials. Because LAuReL has not released its source code, we reimplemented the method based on our understanding of the paper. Detailed training hyperparameters are provided in Table 9.

