# OpenReview forum: "From Transformer to Transponder: Introducing Contextual Modulation Training for Residual Learning  in LLMs"
_ICLR.cc/2026/Conference — Submitted to ICLR 2026_

### Official Review · Reviewer_SMQN · 2025-10-23

**Soundness:** 2
**Presentation:** 1
**Contribution:** 2
**Rating:** 2
**Confidence:** 4

**Summary:**

This paper introduces TRANSPONDER, a method to enhance Transformer-based language models by adding lightweight, context-aware modulators with curvature-controlled sigmoids and low-rank bottlenecks. The approach achieves consistent perplexity reductions across LLaMA backbones (60M–250M parameters) with <1% additional parameters and demonstrates stability in Post-LN settings.

**Strengths:**

It presents a modular framework for input-aware residual pathway control in Transformers.

It demonstrates consistent perplexity improvements across LLaMA variants (60M–250M).

**Weaknesses:**

The core idea of attaching lightweight, contextual gates to existing sublayers is a natural extension of existing gating mechanisms, and as a result, the conceptual novelty is incremental.

It is only evaluated on LLaMA backbones, without exploring its effectiveness on other Transformer variants.

It lacks theoretical analysis to explain why contextual modulation stabilizes training or improves performance.

They do not compare this parameter allocation with alternative ways to use the same parameter budget (e.g., widening FFN or adding lightweight adapters). Therefore, it is unclear whether the observed improvement is specific to the proposed mechanism or simply a result of having more capacity.

**Questions:**

Minor Errors

The baseline perplexity values for the C4 dataset at 130M differ between Table 2 (26.73) and Table 5 (26.07).

The manuscript uses hyphens and en dashes inconsistently for the same purpose. For example, "Self-attention" (Line 128) uses a hyphen, whereas "self–attention" (Line 152) uses an en dash.

The terminology is used inconsistently throughout the manuscript. For example, both “sub-layer” and “sublayer” appear in different sections.

The manuscript is inconsistent in its capitalization of paragraph or subsection titles. For example, “Sigmoid and Learnable Sigmoid.” uses title case, whereas “Hidden dimensions.” uses sentence case.

Lines 156 and 161: Eq. equation 1 -> Eq. 1

Lines 236, 272, 325, 334, and 348: Openwebtext -> OpenWebText

Line 303: !1% -> 1%

Line 610: REPORDUCTION -> REPRODUCTION

Lines 128 and 129: e.g. -> e.g.,

Line 255: LAuREL -> LAuReL

Line 235: corpus -> corpora

Line 236: For OpenWebText dataset -> For the OpenWebText dataset, for C4 dataset -> for the C4 dataset

Line 241: metrics -> metric

Line 402: comparable)perplexity -> comparable) perplexity

Line 587: Learning rate Decay Method -> Learning Rate Decay Method

Line 589: Layer Number -> Number of Layers

Line 590: Head Number -> Number of Heads

---

> ### Author Response · Authors · 2025-12-03
> **Reply to Reviewer SMQN(1/3)**
>
> We appreciate the reviewer's careful reading and valuable comments. We reply to the reviewer’s questions and concerns as follows.
>
> **W1:** The core idea of attaching lightweight, contextual gates to existing sublayers is a natural extension of existing gating mechanisms, and as a result, the conceptual novelty is incremental.
>
> **Reply:** We thank the reviewer for the opportunity to clarify the conceptual differences between Transponder and conventional gating methods. In short, Transponder’s modulators are designed as a two-level meta-learning mechanism: a channel-wise modulator and a global scalar modulator jointly adjust the expression of each linear layer’s computation. This is inspired by how glial cells and epigenetic processes regulate when and how neurons or genes express their activity, rather than changing the underlying wiring or sequence.
>
> Although we initially planned to discuss this bio-inspired meta-learning view in a follow-up journal article, we agree that the distinction is important for the current work. In the revised version, we explicitly describe this perspective and clearly separate Transponder from standard input-aware gating methods such as SDPA [1]. We also update the main text and Figure 1 to make this difference more visible.
>
> To avoid burdening the reviewer with re-reading the entire manuscript, we also provide below a concise summary of this neuro-glia-inspired meta-learning view and its key differences from **gating-based methods** such as SDPA [1].
>
> 1. **Conceptual Difference: Bio-inspired Hierarchical Meta-learning perspective.** As shown in the revised Figure 1, conceptually, our modulators are designed as a bio-inspired hierarchical meta-learning mechanism: the modulators control the expression of each component's computation, analogous to how glial cells or epigenetic mechanisms regulate when and how underlying neurons or genes express their activity. Biologically, glial cells are now recognized as active regulators of information processing rather than passive support cells: they modulate synaptic efficacy, neuronal excitability, and local circuit states in a context-dependent way, providing a flexible “control layer” on top of relatively stable synaptic connectivity [4,5,6]. Thus computational neuroscientists [6,7,8] often frame glia as the optimization layer supervising neural learning. In parallel, epigenetic mechanisms adjust the accessibility and expression level of genes without changing the underlying DNA sequence, enabling long-term, experience-dependent adaptation of gene expression profiles to environmental or developmental context [9,10]. Transponder follows this neuro-glia and epigenetic perspective by decoupling what the backbone computes from how and when these computations are expressed, via lightweight modulators.
>
>    Compared to standard meta-learning in machine learning, which is usually defined over a distribution of tasks and aims to “learn how to learn” across task shifts, our use of meta-learning in Transponder is hierarchical rather than across tasks. In our setting, there is a single main task (e.g., language modeling), but the optimization is decomposed into two coupled levels: 1) The base Transformer layers learn task-specific representations (the usual optimization objective). 2) The Transponder modulators learn how to help the base layers learn, by contextually modulating their activations and gradients. The modulators are not optimized to solve the prediction task directly; instead, they are optimized for their indirect effect on the performance of the base network—i.e., to improve how the base layers are trained and used.
>
>    In this sense, Transponder implements a form of hierarchical meta-learning:
>    the lower level learns the task itself, while the higher level (the modulators) learns how to organize, regulate, and adapt the learning dynamics of the base layers, analogous to “learning how to learn” but applied within a single task rather than across multiple tasks.
>
> 2. **Where the modulator acts.** SDPA [1] does not modify the internal attention computation; instead, it applies a gate only to the attention output, using activations after RMSNorm. Transponder, in contrast, attaches modulators to all linear layers (including attention and MLP projections). This constructs an explicit meta-learning layer that controls information scaling throughout the block, rather than a single gate at the attention output.
>
> 3. **Granularity of modulation.** For each linear layer, Transponder uses two complementary modulators. The channel-wise modulator provides a fine-grained controller that selectively amplifies or attenuates individual feature channels, while the global scalar modulator provides a coarse, layer-level controller that scales the overall contribution of the transformation. Together, they form a lightweight, bio-inspired meta-learner that modulates computations across layers as a function of context, instead of acting as a single gating.

---

> ### Author Response · Authors · 2025-12-03
> **Reply to Reviewer SMQN(2/3)**
>
> We want to thank the reviewer for the opportunity to clarify the motivation of introducing Transponder. Although we initially planned to discuss this bio-inspired meta-learning view in a follow-up journal article, we agree that the distinction is important for the current work. In the revised version, we explicitly describe this perspective and clearly separate Transponder from standard input-aware gating methods such as SDPA [1]. We also update the main text and Figure 1 to make this clarification more visible.
>
> **W2:** It is only evaluated on LLaMA backbones, without exploring its effectiveness on other Transformer variants.
>
> **Reply:** Thank you for the constructive question. To verify that Transponder is not restricted to LLaMA-style architectures, we additionally evaluate it on two different Transformer families:
>
> 1) Transformer-Base (Encoder–Decoder) on three machine translation benchmarks (Multi30K, IWSLT, WMT). The results are averaged under 3 seeds and the standard error is reported.
>
> Transponder consistently improves BLEU across all datasets, as shown below:
>
> | Model             | Multi30K (BLEU)     | IWSLT (BLEU)        | WMT (BLEU) |
> |-------------------|----------------------|----------------------|------------|
> | Transformer-Base  | 31.38 ± 0.38         | 24.48 ± 0.30         | 25.22      |
> | Transponder        | **33.03 ± 0.29**     | **25.45 ± 0.28**     | **26.08**  |
>
>
> 2. OPT-125M (Decoder-Only, distinct from LLaMA) on language modeling.
> Transponder also shows consistent improvements in perplexity:
>
> | Model           | PPL ↓ |
> |----------------|--------|
> | Widen-Original | 25.87  |
> | SDPA    | 24.97  |
> | Transponder    | **23.58** |
>
> These results demonstrate that Transponder generalizes beyond LLaMA and remains effective across encoder–decoder models, decoder-only models, and diverse tasks (machine translation + language modeling).
>
> **W3:** It lacks theoretical analysis to explain why contextual modulation stabilizes training or improves performance.
>
> **Reply:**
>
> 1. **Stabilizing training via smoother gradients.**
>    In the new **Fig. 2** we plot the per-step input gradient norm (averaged over all 7 decoder layers) for LLaMA-60M trained for 10k steps with the three variants: baseline, SDPA, and Transponder. While all methods share a short warm-up at the beginning, the baseline and SDPA models exhibit frequent and large gradient spikes throughout training, indicating that they are much more sensitive to the learning rate and prone to instability. In contrast, Transponder keeps the gradient norms low and smooth over the entire training trajectory. This directly shows that contextual modulation acts as a stabilizer of the optimization dynamics, allowing us to safely use higher learning rates without divergence.
>
>    To demonstrate this, we conduct a learning rate sweep on the LLaMA60M and LLaMA130M model. These results demonstrate that Transponder is substantially more robust to large learning rates, and its performance can even benefit from a more aggressive learning-rate regime—unlike the baseline or SDPA, which both exhibit strong sensitivity and training failures under the same conditions. We thank the Reviewer for giving us the oppotunity to think deeper the reason behind the advantage and we have updated the results in the revised manuscript using a higher learning rate for Transponder. Below the LLaMA_widen is the parameter-matched baseline by widening the FFN intermediate dimension such that the parameter increase is approximately the same as the Transponder.
>
>     | Model Size | Dataset     | Model       | 1e-2    | 3e-3   | 1e-3  |
>     | -| -| - | -| -| ----- |
>     | **60M**    | OWT         | LLaMA_widen    | 1007.00 | 600.88  | 26.59 |
>     |            |             | SDPA        | 524.93  | 22.79  | 25.18 |
>     |            |             | Transponder | **22.07**   | 22.65  | 25.85 |
>     | **60M**    | C4          | LLaMA_widen    | 986.52  | 540.23  | 30.05 |
>     |            |             | SDPA        | 693.65  | 29.24  | 31.84 |
>     |            |             | Transponder | **28.06**   | 28.53  | 32.24 |
>     | **130M**   | OWT | LLaMA_widen    | 683.08  | 632.76  | 19.74 |
>     |            |             | SDPA        | 610.09  | 20.44  | 18.60 |
>     |            |             | Transponder | 18.49   | **17.45** | 18.26 |
>     | **130M**   | C4          | LLaMA_widen    | 775.34  | 669.47  | 26.63 |
>     |            |             | SDPA        | 724.76  | 302.44 | 23.92 |
>     |            |             | Transponder | **21.82**   | 22.13  | 23.99 |
>
>     At high learning rates (1e-2 and 3e-3), the widened baseline often collapses to extremely high PPL, and SDPA also degrades sharply. Transponder, however, remains stable and in several cases achieves its best performance at a higher learning rate, which would not be usable for the baseline. This robustness to large learning rates is directly consistent with the smoother gradient behavior observed in Fig. 2.

---

> ### Author Response · Authors · 2025-12-03
> **Reply to Reviewer SMQN(3/3)**
>
> 2. Beyond stability, the three design choices highlighted above also contribute to the improvement: 1) A meta-learning perspective, where modulators learn how strongly each computation should be expressed rather than directly changing the underlying weights. 2) An all-layer modulation scheme, where all the linear layers in attention and MLP projections are consistently modulated rather than only gating the attention output. 3) A two-level modulator (channel-wise + scalar), which provides both fine-grained and coarse control over the representation.
>
>
>
>
> **W4:** They do not compare this parameter allocation with alternative ways to use the same parameter budget (e.g., widening FFN or adding lightweight adapters). Therefore, it is unclear whether the observed improvement is specific to the proposed mechanism or simply a result of having more capacity.
>
> **Reply:** Thank you for this insightful suggestion. To ensure a fully fair comparison, we trained a parameter-matched baseline by widening the FFN intermediate dimension such that the parameter increase is approximately the same as Transponder. We denote this model as LLaMA_widen.
>
> Below we report the updated results for the four variants—LLaMA (baseline), LLaMA_widen, SDPA, and Transponder—across 60M, 130M, and 250M model scales. We keep this table in the main paper and move the remaining ablations to the Appendix Table 4 to save space.
>
> | Datasets    | Model            | LLaMA60M  | LLaMA130M | LLaMA250M |
> | ----------- | ---------------- | --------- | --------- | --------- |
> | Openwebtext | LLaMA (baseline) | 26.56     | 19.27     | 17.28     |
> |             | LLaMA_widen      | 26.59     | 19.74     | 17.10     |
> |             | SDPA             | 22.79     | 20.44     | 15.65     |
> |             | Transponder      | **22.07** | **17.45** | **14.93** |
> | C4          | LLaMA (baseline) | 30.31     | 26.73     | 21.92     |
> |             | LLaMA_widen      | 30.28     | 26.63     | 21.79     |
> |             | SDPA             | 29.24     | 23.92     | 19.12     |
> |             | Transponder      | **28.06** | **21.82** | **18.72** |
>
> We additionally evaluate the same four models at the 1B parameter scale on OpenWebText. These results are included in the Table 1 for completeness.
>
> | Model            | LLaMA1B |
> | ---------------- | ------- |
> | LLaMA (baseline) | 14.63   |
> | LLaMA_widen      | 14.55   |
> | SDPA             | 13.82   |
> | Transponder      | **12.51**|
>
> Across all model sizes (60M → 1B) and datasets (OpenWebText, C4), the widened baseline (+1% params) performs very similarly to the original baseline, confirming that simple parameter scaling alone does not yield meaningful perplexity gains as Transponder. In contrast, Transponder consistently improves perplexity at all scales, demonstrating that its benefits come from its contextual modulation mechanism rather than from extra parameters.
>
> **Minor Errors:**
>
> **Reply:**  Thanks for pointing out all these typos and mistakes. We have addressed all these points in the updated manuscript.
>
> [1] Zihan Qiu. Gated Attention for Large Language Models: Non-linearity, Sparsity, and Attention-Sink-Free. NeurIPS 2025.
>
> [2] Fields, R. D. (2004). The other half of the brain. Scientific American.
>
> [3] Araque, A., Carmignoto, G., Haydon, P. G., Oliet, S. H. R., Robitaille, R., and Volterra, A. (2014). Gliotransmitters travel in time and space. Neuron.
>
> [4] De Pittà, M., and Brunel, N. (2016). Astrocytes: orchestrating synaptic plasticity? Neuroscience.
>
> [5] De Pittà et al., 2011. A Tale of Two Stories: Astrocyte Regulation of Synaptic Depression and Facilitation. PLoS Computational Biology.
>
> [6] Volman et al., 2012. Computational models of neuron–astrocyte interaction in synaptic transmission. Frontiers in Computational Neuroscience.
>
> [7] Bird, A. (2007). Perceptions of epigenetics. Nature.
>
> [8] Sweatt, J. D. (2013). The emerging field of neuroepigenetics. Neuron.

---

### Official Review · Reviewer_Kpr8 · 2025-10-31

**Soundness:** 3
**Presentation:** 3
**Contribution:** 3
**Rating:** 4
**Confidence:** 3

**Summary:**

This paper proposes TRANSPONDER, a lightweight context-aware modulation framework for Transformers that dynamically scales the outputs of sublayers (e.g., Q/K/V projections, FFN) according to input context. The method separates representation learning (“what to compute”) from residual control (“how much to mix”), introducing compact modulators that operate at token- and channel-level granularity with <1% additional parameters. Experiments on LLaMA models (60M-250M) show consistent 5-15% perplexity reductions on OpenWebText and C4 datasets and improved training stability in the Post-LN setting. The paper includes ablations over placement, granularity, and hidden dimension.

**Strengths:**

- Addresses the static nature of residual scaling in Transformers.
- Can be integrated into standard architectures with minimal modification.
- 5-15% perplexity reductions across datasets and model sizes.
- Prevents divergence in challenging Post-LN setups.
- Explores modulation placement, granularity, and hidden size.
- Visual analyses show token- and depth-dependent scaling patterns.
- Achieves improvements with <1% additional parameters.

**Weaknesses:**

- Results are limited to language modeling on OpenWebText/C4 using 60M-250M models. No evidence on large-scale models (>=1B) or real downstream tasks (QA, reasoning, instruction following).
- Claims of “lightweight” and “negligible overhead” are not supported by FLOPs, latency, or memory statistics.
- No analysis of how contextual modulation stabilizes training or enhances representational capacity.
- No comparison against equal-parameter alternatives such as wider FFNs or more attention heads that could yield similar gains.
- Partial overlap with LLaMA’s built-in gate projections: A single ablation (“w/o up and gate”) hints at redundancy but lacks a full quantitative study.

**Questions:**

- Could you provide runtime, FLOPs, and memory overhead compared to LLaMA baselines across different sequence lengths and batch sizes?
- How does TRANSPONDER perform on larger-scale models (>=1B) and downstream tasks such as instruction following or reasoning benchmarks?
- If the same +1% parameter budget were spent on FFN expansion or additional attention heads, would performance improvements be comparable?
- Can you offer a theoretical or analytical explanation for why contextual modulation improves optimization and stabilizes Post-LN training?
- Have you analyzed the interaction or redundancy between your modulators and LLaMA’s gate-proj components?

---

> ### Author Response · Authors · 2025-12-03
> **Reply to Reviewer Kpr8(1/3)**
>
> We thank the reviewer for the thoughtful review, positive assessment, and helpful critical remarks. We address all points raised by the reviewer in the following.
>
> **W1:** Results are limited to language modeling on OpenWebText/C4 using 60M-250M models. No evidence on large-scale models (>=1B) or real downstream tasks (QA, reasoning, instruction following).
>
> **Reply:** To address this concern, we further evaluate Transponder on a larger 1B-parameter LLaMA model, trained for 100K steps under the same training setup. The results show that Transponder continues to deliver substantial perplexity improvements at this larger scale:
>
> - Original Baseline: 14.63
> - SDFA: 13.87
> - Transponder: **12.56**
>
> These results demonstrate that Transponder’s gains persist to ≥1B-scale architectures.
>
> To further evaluate whether the improvements go beyond perplexity, we performed zero-shot downstream evaluation on GLUE and MMLU using accuracy. To improve generalization for all models in a fair manner, we extended the training to 60K steps (≈11.8B tokens) and increased max sequence length to 1024 for LLaMA 60M. The results are summarized in the table below, showing that Transponder achieves the highest average accuracy and the highest win-rate (6/10 tasks), demonstrating consistent gains beyond perplexity. Below the LLaMA_widen is the parameter-matched baseline by widening the FFN intermediate dimension such that the parameter increase is approximately the same as the Transponder.
>
> | Dataset          | SDPA | Transponder | LLaMA_widen |
> |------------------|------------|-------------|----------------|
> | **CoLA (mcc)**   | 0.0000     | **0.0464**  | 0.0230         |
> | **COPA (acc)**   | 0.56       | **0.61**    | 0.58           |
> | **HellaSwag_norm** | 0.2989   | **0.3036**  | 0.2928         |
> | **MNLI (acc)**   | **0.3281** | 0.3264      | 0.3277         |
> | **MRPC (acc)**   | **0.6912** | 0.6838      | 0.6201         |
> | **QNLI (acc)**   | 0.4944     | **0.4946**  | 0.4937         |
> | **QQP (acc)**    | 0.3683     | 0.3685      | **0.3691**     |
> | **RTE (acc)**    | **0.5343** | 0.5307      | 0.4765         |
> | **SST-2 (acc)**  | 0.4908     | **0.5195**  | 0.4908         |
> | **MMLU (acc)**   | 0.22953    | **0.23002** | 0.22924        |
> | **Average (9 tasks)** | 0.4184 | **0.4315** | 0.4082         |
> | **Win-Rate (10 tasks)** | 3 / 10 | **6 / 10** | 1 / 10       |
>
> **W2:** Claims of “lightweight” and “negligible overhead” are not supported by FLOPs, latency, or memory statistics.
>
> **Reply:** Thank you for the comment. We now provide FLOPs and latency measurements to support our claim that Transponder adds only a small overhead. All numbers are obtained on LLaMA-60M evaluated on a single A100 GPU.
>
> **Inference Latency (Throughput, seq_len = 256)**
>
> | Model        | BS=32      | BS=64      | BS=128     |
> | ------------ | ---------- | ---------- | ---------- |
> | **LLaMA_widen** | **707.57** | **721.81** | **748.74** |
> | SDPA   | 672.98     | 688.83     | 707.66     |
> | Transponder  | 654.53     | 658.96     | 689.76     |
>
> **Per-token FLOPs**
>
> | Model       | Forward FLOPs/token (GFLOPs) | Training FLOPs/token (GFLOPs) |
> | -| -| ----------------------------- |
> | **LLaMA_widen** | 0.082                        | 0.245                         |
> | SDPA  | 0.088                        | 0.264                         |
> | Transponder | 0.084                        | 0.252                         |
>
> Overall, the results show that Transponder adds a small compute and latency cost. The FLOPs increase is about 2–3%, and the inference throughput decreases by around 7–9% depending on batch size. This overhead is smaller than that of the SDPA baseline and stays modest relative to the improvements achieved.
>
> **W3:** No analysis of how contextual modulation stabilizes training or enhances representational capacity.
>
> **Reply:** We thank the reviewer for this comment. We appreciate this question and agree that the magnitude of the improvement merits explanation. The performance gain mainly comes from more stable optimization, which allows us to exploit higher learning rates, and from the meta-learning style modulation introduced by Transponder.
>
> 1. **Stabilizing training via smoother gradients.**
>    In the new **Fig. 2** we plot the per-step input gradient norm (averaged over all 7 decoder layers) for LLaMA-60M trained for 10k steps with the three variants: baseline, SDPA, and **Transponder**. While all methods share a short warm-up at the beginning, the baseline and SDPA models exhibit frequent and large gradient spikes throughout training, indicating that they are much more sensitive to the learning rate and prone to instability. In contrast, Transponder keeps the gradient norms low and smooth over the entire training trajectory. This directly shows that contextual modulation acts as a stabilizer of the optimization dynamics, allowing us to safely use higher learning rates without divergence.

---

> ### Author Response · Authors · 2025-12-03
> **Reply to Reviewer Kpr8(2/3)**
>
> To demonstrate this, we conduct a learning rate sweep on the LLaMA60M and LLaMA130M model. These results demonstrate that Transponder is substantially more robust to large learning rates, and its performance can even benefit from a more aggressive learning-rate regime—unlike the baseline or SDPA, which both exhibit strong sensitivity and training failures under the same conditions. We thank the Reviewer for giving us the oppotunity to think deeper the reason behind the advantage and we have updated the results in the revised manuscript using a higher learning rate for Transponder. Below the LLaMA_widen is the parameter-matched baseline by widening the FFN intermediate dimension such that the parameter increase is approximately the same as the Transponder.
>
> | Model Size | Dataset     | Model       | 1e-2    | 3e-3   | 1e-3  |
> | ---------- | ----------- | ----------- | ------- | ------ | ----- |
> | **60M**    | OWT         | LLaMA_widen    | 1007.00 | 600.88  | 26.59 |
> |            |             | SDPA        | 524.93  | 22.79  | 25.18 |
> |            |             | Transponder | **22.07**   | 22.65  | 25.85 |
> | **60M**    | C4          | LLaMA_widen    | 986.52  | 540.23  | 30.05 |
> |            |             | SDPA        | 693.65  | 29.24  | 31.84 |
> |            |             | Transponder | **28.06**   | 28.53  | 32.24 |
> | **130M**   | OWT | LLaMA_widen    | 683.08  | 632.76  | 19.74 |
> |            |             | SDPA        | 610.09  | 20.44  | 18.60 |
> |            |             | Transponder | 18.49   | **17.45** | 18.26 |
> | **130M**   | C4          | LLaMA_widen    | 775.34  | 669.47  | 26.63 |
> |            |             | SDPA        | 724.76  | 302.44 | 23.92 |
> |            |             | Transponder | **21.82**   | 22.13  | 23.99 |
>
> At high learning rates (1e-2 and 3e-3), the widened baseline often collapses to extremely high PPL, and SDPA also degrades sharply. Transponder, however, remains stable and in several cases achieves its best performance at a higher learning rate, which would not be usable for the baseline. This robustness to large learning rates is directly consistent with the smoother gradient behavior observed in Fig. 2.
>
> 2. **Effect of the modulator design itself.**
> Beyond stability, the representational capacity is also shown by the above test of zero-shot evaluation that suggested by the Reviewer. Three design choices highlighted above also contribute to the representation: 1) The hirechical meta-learning perspective, where modulators learn how strongly each computation should be expressed rather than directly changing the underlying weights. 2) The all-layer modulation scheme, where all the linear layers in attention and MLP projections are consistently modulated rather than only gating the attention output. 3) The two-level modulator (channel-wise + scalar), which provides both fine-grained and coarse control over the representation.
>
>
> **W4:** No comparison against equal-parameter alternatives such as wider FFNs or more attention heads that could yield similar gains.
>
> **Reply:** Thank you for this insightful suggestion. To ensure a fully fair comparison, we trained a parameter-matched baseline by widening the FFN intermediate dimension such that the parameter increase is approximately +1%, comparable to the overhead introduced by Transponder. We denote this model as LLaMA_widen.
>
> Below we report the updated results for the four variants—LLaMA (baseline), LLaMA_widen, SDPA, and Transponder—across 60M, 130M, and 250M model scales. We keep this table in the main paper and move the remaining ablations to the Appendix Table 10 to save space.
>
> | Datasets    | Model            | LLaMA60M  | LLaMA130M | LLaMA250M |
> | - | -| -| -| -|
> | Openwebtext | LLaMA (baseline) | 26.56     | 19.27     | 17.28     |
> | | LLaMA_widen      | 26.59     | 19.74     | 17.10     |
> | | SDPA             | 22.79     | 20.44     | 15.65     |
> |  | Transponder      | **22.07** | **17.45** | **14.93** |
> | C4          | LLaMA (baseline) | 30.31     | 26.73     | 21.92     |
> |             | LLaMA_widen      | 30.28     | 26.63     | 21.79     |
> |             | SDPA             | 29.24     | 23.92     | 19.12     |
> |             | Transponder      | **28.06** | **21.82** | **18.72** |
>
> We additionally evaluate the same four models at the 1B parameter scale on OpenWebText. These results are included in the Table 1 for completeness.
>
> | Model            | LLaMA1B |
> | --| -|
> | LLaMA (baseline) | 14.63   |
> | LLaMA_widen      | 14.55   |
> | SDPA             | 13.82   |
> | Transponder      | **12.51**|
>
> Across all model sizes (60M → 1B) and datasets (OpenWebText, C4), the widened baseline (+1% params) performs very similarly to the original baseline, confirming that simple parameter scaling alone does not yield meaningful perplexity gains. In contrast, Transponder consistently improves perplexity at all scales, demonstrating that its benefits come from its contextual modulation mechanism rather than from extra parameters.

---

> ### Author Response · Authors · 2025-12-03
> **Reply to Reviewer Kpr8(3/3)**
>
> **W5:** Partial overlap with LLaMA’s built-in gate projections: A single ablation (“w/o up and gate”) hints at redundancy but lacks a full quantitative study.
>
> **Reply:** Thanks for this comment. We agree that a more systematic analysis is needed. To isolate the contributions of (1) LLaMA’s native gate_proj and (2) our modulators, we conduct a full ablation over four configurations: Gate only (LLaMA-baseline), Modulator only, Both, and Neither. Results on LLaMA-60M and LLaMA-130M are shown below:
>
> | Setting                    | LLaMA-60M | LLaMA-130M |
> |---------------------------|-----------|------------|
> | w/o gate and modulators   | 32.61     | 27.95      |
> | w only gate               | 30.31     | 26.73      |
> | w only modulators         | 30.14     | 26.48      |
> | w both                    | **29.91** | **26.26**  |
>
> 1) Both mechanisms together achieve the best perplexity.
>
> 2) Modulator-only consistently outperforms gate-only, indicating that our low-rank channel-wise modulator + layer-wise scalar modulator provide complementary benefits beyond the built-in gating.
>
> 3) Removing both gives the worst performance.
> We have conducted a more quantitative study of the difference of the gate_proj and the modulator.
> Gate + Modulator, Gate only, Modulator only, Without both.
>
> To further rule out any architectural coupling with LLaMA’s gate_proj, we also evaluate on OPT-125M, which contains no FFN gate_proj. In this setting, Transponder still achieves the best perplexity (23.58), outperforming both the widened baseline and SDPA. This confirms that the benefits of Transponder do not rely on LLaMA’s inherent gating mechanism.
>
> | Model           | PPL ↓ |
> |----------------|--------|
> | Widen-Original | 25.87  |
> | SDPA     | 24.97  |
> | Transponder    | **23.58** |

---

### Official Review · Reviewer_7pnL · 2025-10-31

**Soundness:** 2
**Presentation:** 3
**Contribution:** 2
**Rating:** 4
**Confidence:** 3

**Summary:**

This paper proposes Transponder, a lightweight, input-aware modulation mechanism for Transformers. The key idea is to add contextual modulators that scale outputs of linear or functional sublayers (e.g., attention, MLP) at token or channel level. This allows the model to regulate residual information flow dynamically, unlike static scaling methods such as ReZero, DeepNorm, or LAuReL.
Experiments on LLaMA backbones (60M–250M) trained on OpenWebText and C4 show consistent perplexity (PPL) improvements with less than 1% parameter overhead. The authors also perform ablations on granularity, placement, and modulation strength, and visualize token- and depth-dependent modulator behaviors.

**Strengths:**

1. The idea of decoupling representation from control via input-dependent modulators is conceptually strong and intuitively motivated.
2. Experiments are strong. Covers multiple LLaMA scales and extensive ablations (modulator placement, resolution, hidden dimension, and component-wise contribution).

**Weaknesses:**

1. Similarity to Gating Methods. Perhaps a discussion comparing it with the gating method can be seen during the rebuttal phase.
2. In Table 1, the first-row results (“Modulator-path-scalar”) show abnormally high PPL (e.g., 1088 for 250M), suggesting instability. The authors should explain why this configuration fails and whether this is due to optimization divergence or implementation bugs.
3. Table 1 lacks direct comparison with the LLaMA baseline for those variants, making it hard to gauge how much each modulator improves over standard training.

**Questions:**

I would like to ask about the significant reduction in PPL for Table 4. What is the reason behind this performance improvement? The method mentioned in the paper still follows a Transformer-like architecture, so theoretically, with the same number of parameters, there shouldn't be such a substantial performance change.

---

> ### Author Response · Authors · 2025-12-03
> **Reply to Reviewer 7pnL(1/3)**
>
> We are grateful to the reviewer for the thorough evaluation and many insightful suggestions, which helped us improve the clarity and completeness of the paper. We respond to each comment as follows.
>
> **W1:** Similarity to Gating Methods. Perhaps a discussion comparing it with the gating method can be seen during the rebuttal phase.
>
> **Reply:** We thank the reviewer for the opportunity to clarify the conceptual differences between Transponder and conventional gating methods. In short, Transponder’s modulators are designed as a two-level meta-learning mechanism: a channel-wise modulator and a global scalar modulator jointly adjust the expression of each linear layer’s computation. This is inspired by how glial cells and epigenetic processes regulate when and how neurons or genes express their activity, rather than changing the underlying wiring or sequence.
>
> Although we initially planned to discuss this bio-inspired meta-learning view in a follow-up journal article, we agree that the distinction is important for the current work. In the revised version, we explicitly describe this perspective and clearly separate Transponder from standard input-aware gating methods such as SDPA [1]. We also update the main text and Figure 1 to make this difference more visible.
>
> To avoid burdening the reviewer with re-reading the entire manuscript, we also provide below a concise summary of this **neuro-glia-inspired meta-learning view** and its key differences from **gating-based methods** such as SDPA [1].
>
> 1. **Conceptual Difference: Bio-inspired Hierarchical Meta-learning perspective.** As shown in the revised Figure 1, conceptually, our modulators are designed as a bio-inspired hirechical meta-learning mechanism: the modulators control the expression of each component's computation, analogous to how glial cells or epigenetic mechanisms regulate when and how underlying neurons or genes express their activity. Biologically, glial cells are now recognized as active regulators of information processing rather than passive support cells: they modulate synaptic efficacy, neuronal excitability, and local circuit states in a context-dependent way, providing a flexible “control layer” on top of relatively stable synaptic connectivity [4,5,6]. Thus computational neuroscientists [6,7,8] often frame glia as the optimization layer supervising neural learning. In parallel, epigenetic mechanisms adjust the accessibility and expression level of genes without changing the underlying DNA sequence, enabling long-term, experience-dependent adaptation of gene expression profiles to environmental or developmental context [9,10]. Transponder follows this neuro-glia and epigenetic perspective by decoupling what the backbone computes from how and when these computations are expressed, via lightweight modulators.
>
>    Compared to standard meta-learning in machine learning, which is usually defined over a distribution of tasks and aims to “learn how to learn” across task shifts, our use of meta-learning in Transponder is hierarchical rather than across tasks. In our setting, there is a single main task (e.g., language modeling), but the optimization is decomposed into two coupled levels: 1) The base Transformer layers learn task-specific representations (the usual optimization objective). 2) The Transponder modulators learn how to help the base layers learn, by contextually modulating their activations and gradients. The modulators are not optimized to solve the prediction task directly; instead, they are optimized for their indirect effect on the performance of the base network—i.e., to improve how the base layers are trained and used.
>
>    In this sense, Transponder implements a form of hierarchical meta-learning:
>    the lower level learns the task itself, while the higher level (the modulators) learns how to organize, regulate, and adapt the learning dynamics of the base layers, analogous to “learning how to learn” but applied within a single task rather than across multiple tasks.
>
> 2. **Where the modulator acts.** SDPA [1] does not modify the internal attention computation; instead, it applies a gate only to the attention output, using activations after RMSNorm. Transponder, in contrast, attaches modulators to all linear layers (including attention and MLP projections). This constructs an explicit meta-learning layer that controls information scaling throughout the block, rather than a single gate at the attention output.

---

> ### Author Response · Authors · 2025-12-03
> **Reply to Reviewer 7pnL(2/3)**
>
> 3. **Granularity of modulation.** For each linear layer, Transponder uses two complementary modulators. The channel-wise modulator provides a fine-grained controller that selectively amplifies or attenuates individual feature channels, while the global scalar modulator provides a coarse, layer-level controller that scales the overall contribution of the transformation. Together, they form a lightweight, bio-inspired meta-learner that modulates computations across layers as a function of context, instead of acting as a single gating.
>
> We have added a dedicated paragraph in the Appendix C that explicitly compares Transponder to gating-based methods (such as SDPA [1]) and highlights this bio-inspired meta-learning role.
>
> **W2:** In Table 1, the first-row results (“Modulator-path-scalar”) show abnormally high PPL (e.g., 1088 for 250M), suggesting instability. The authors should explain why this configuration fails and whether this is due to optimization divergence or implementation bugs.
>
> **Reply:** Thank you for pointing this out. As you noted for 250M, we also observe that on LLaMA-60M the “Modulator-path-scalar” configuration reaches a PPL of 62.91, which is clearly abnormal compared to the baseline and the other variants.
>
> In Table 1 we purposely use exactly the same hyperparameters (including the learning rate) for all variants in order to isolate architectural effects. In particular, all models there are trained with a learning rate of 3e-3. However, as shown in the sensitivity study (Table 10 in the Appendix), this learning rate is already too aggressive for both the LLaMA baseline and SDPA, which often fail or degrade severely under this setting.
>
> For “Modulator-path-scalar” the situation is even more fragile: a single scalar controls the entire functional path, so under a large learning rate its gradients become very noisy and conflicting. This tends to push the scalar into saturation and leads to poor use of depth and capacity, resulting in the extremely high PPLs. By contrast, the layer-scalar and layer-channel variants introduce finer-grained modulators at each linear layer, which distribute the learning signal more evenly and yield much smoother gradient norms (see Fig. 2). This finer-grained control is also one of the reasons why the full Transponder configuration can be trained stably even at relatively high learning rates.
>
> **W3:** Table 1 lacks direct comparison with the LLaMA baseline for those variants, making it hard to gauge how much each modulator improves over standard training.
>
> **Reply:** Thanks for this advice. In the revised version, we have added the corresponding LLaMA baseline rows into Table 1 as well.
>
> **Q1:** I would like to ask about the significant reduction in PPL for Table 4. What is the reason behind this performance improvement? The method mentioned in the paper still follows a Transformer-like architecture, so theoretically, with the same number of parameters, there shouldn't be such a substantial performance change.
>
> **Reply:** We appreciate this question and agree that the magnitude of the improvement merits explanation. The performance gain mainly comes from more stable optimization, which allows us to exploit higher learning rates, and from the meta-learning style modulation introduced by Transponder.
>
> 1. **Stabilizing training via smoother gradients.**
>    In the new **Fig. 2** we plot the per-step input gradient norm (averaged over all 7 decoder layers) for LLaMA-60M trained for 10k steps with the three variants: baseline, SDPA, and **Transponder**. While all methods share a short warm-up at the beginning, the baseline and SDPA models exhibit frequent and large gradient spikes throughout training, indicating that they are much more sensitive to the learning rate and prone to instability. In contrast, Transponder keeps the gradient norms low and smooth over the entire training trajectory. This directly shows that contextual modulation acts as a stabilizer of the optimization dynamics, allowing us to safely use higher learning rates without divergence.
>
>    To demonstrate this, we conduct a learning rate sweep on the LLaMA60M and LLaMA130M model. These results demonstrate that Transponder is substantially more robust to large learning rates, and its performance can even benefit from a more aggressive learning-rate regime—unlike the baseline or SDPA, which both exhibit strong sensitivity and training failures under the same conditions. We thank the Reviewer for giving us the oppotunity to think deeper the reason behind the advantage and we have updated the results in the revised manuscript using a higher learning rate for Transponder. Below the LLaMA_widen is the parameter-matched baseline by widening the FFN intermediate dimension such that the parameter increase is approximately the same as the Transponder.

---

> ### Author Response · Authors · 2025-12-03
> **Reply to Reviewer 7pnL(3/3)**
>
> | Model Size | Dataset     | Model       | 1e-2    | 3e-3   | 1e-3  |
> | ---------- | ----------- | ----------- | ------- | ------ | ----- |
> | **60M**    | OWT         | LLaMA_widen    | 1007.00 | 600.88  | 26.59 |
> |            |             | SDPA        | 524.93  | 22.79  | 25.18 |
> |            |             | Transponder | **22.07**   | 22.65  | 25.85 |
> | **60M**    | C4          | LLaMA_widen    | 986.52  | 540.23  | 30.05 |
> |            |             | SDPA        | 693.65  | 29.24  | 31.84 |
> |            |             | Transponder | **28.06**   | 28.53  | 32.24 |
> | **130M**   | OWT | LLaMA_widen    | 683.08  | 632.76  | 19.74 |
> |            |             | SDPA        | 610.09  | 20.44  | 18.60 |
> |            |             | Transponder | 18.49   | **17.45** | 18.26 |
> | **130M**   | C4          | LLaMA_widen    | 775.34  | 669.47  | 26.63 |
> |            |             | SDPA        | 724.76  | 302.44 | 23.92 |
> |            |             | Transponder | **21.82**   | 22.13  | 23.99 |
>
> At high learning rates (1e-2 and 3e-3), the widened baseline often collapses to extremely high PPL, and SDPA also degrades sharply. Transponder, however, remains stable and in several cases achieves its best performance at a higher learning rate, which would not be usable for the baseline. This robustness to large learning rates is directly consistent with the smoother gradient behavior observed in Fig. 2.
>
> 2. **Effect of the modulator design itself.**
> Beyond stability, the three design choices highlighted above also contribute to the improvement: 1) A meta-learning perspective, where modulators learn how strongly each computation should be expressed rather than directly changing the underlying weights. 2) An all-layer modulation scheme, where all the linear layers in attention and MLP projections are consistently modulated rather than only gating the attention output. 3) A two-level modulator (channel-wise + scalar), which provides both fine-grained and coarse control over the representation.
>
> [1] Zihan Qiu. Gated Attention for Large Language Models: Non-linearity, Sparsity, and Attention-Sink-Free. NeurIPS 2025.
>
> [2] Fields, R. D. (2004). The other half of the brain. Scientific American.
>
> [3] Araque, A., Carmignoto, G., Haydon, P. G., Oliet, S. H. R., Robitaille, R., and Volterra, A. (2014). Gliotransmitters travel in time and space. Neuron.
>
> [4] De Pittà, M., and Brunel, N. (2016). Astrocytes: orchestrating synaptic plasticity? Neuroscience.
>
> [5] De Pittà et al., 2011. A Tale of Two Stories: Astrocyte Regulation of Synaptic Depression and Facilitation. PLoS Computational Biology.
>
> [6] Volman et al., 2012. Computational models of neuron–astrocyte interaction in synaptic transmission. Frontiers in Computational Neuroscience.
>
> [7] Bird, A. (2007). Perceptions of epigenetics. Nature.
>
> [8] Sweatt, J. D. (2013). The emerging field of neuroepigenetics. Neuron.

---

### Official Review · Reviewer_SG6Y · 2025-11-01

**Soundness:** 2
**Presentation:** 3
**Contribution:** 2
**Rating:** 6
**Confidence:** 3

**Summary:**

The paper introduces Transponder, which aims to improve the performance of Transformer-based models by introducing contextual modulation training for residual learning in large language models (LLMs). The authors argue that the static design of Transformers neglects context-sensitive regulation of information flow through residual pathways, and propose the use of lightweight, input-aware modulators to scale the outputs of linear sublayers within a block or the entire block output at token- and channel-level granularity.

Transponder provides improvement over six other scaling or normalization methods across LLaMA backbones ranging from 60M to 250M parameters, yielding consistent perplexity reductions with less than 1% additional parameters. Analysis of learned modulator values reveals depth-, module-, and token-specific patterns that adapt layer-wise contributions to input semantics, providing direct evidence that residual functional transformations benefit from adaptive, context-aware scaling. Transponder provides a simple, general mechanism to augment Transformer-based models with context-sensitive modulators, providing robust and significant performance improvements without substantial architectural changes.

**Strengths:**

* **Well-Motivated Problem**: The authors clearly identify a limitation of the transformer design—static functional scaling across residual pathways—and provide a rationale for introducing input-aware modulation. The problem is convincingly motivated, with emphasis on the need for adaptive regulation to improve representation learning.

* **Clear Writing**: The paper is written in an accessible and structured manner. It carefully explains the core principles of Trasnponder, its mechanism, and its integration into Transformers, making it easy to understand.

* **Strong Empirical Results**: Transponder demonstrates consistent and significant gains in perplexity reduction across LLaMA model variants, with relative improvements reaching as high as 15.3%, underscoring the effectiveness of the approach without substantial computational or parameter overhead.

* **Comprehensive Analysis and Ablations**: The paper includes extensive ablations and experiments, systematically analyzing placement, granularity, hidden sizes, and modulation coverage. This depth of analysis confirms the robustness and adaptability of the design choices.

**Weaknesses:**

* **Evaluation** : It should be possible to train a llama baseline like the one in the paper to achieve less than 22.50 ppl on OWT. Did the authors properly tune the baseline ? What is the experiment setup  ? How many FLOPS are used for the baseline vs the Transponder results ? What is held constant ? It would also be interesting to know if this works beyond ppl and if it works on downstream taks.

* **Efficiency** : How does the Transponder affect the training & inference latency and throughput ? Do these make the models slower ?

**Questions:**

See Weaknesses

---

> ### Author Response · Authors · 2025-12-03
> **Reply to Reviewer SG6Y(1/2)**
>
> We thank the reviewer for carefully reading our paper and for the thoughtful, constructive feedback. We address the reviewer’s comments and questions as follows:
>
> **W1:** Evaluation : It should be possible to train a llama baseline like the one in the paper to achieve less than 22.50 ppl on OWT.
>
> **Reply:** We thank the reviewer for the helpful comment. To verify the claim, we extended the training of the LLaMA-60M baseline and Transponder to 60K steps (7.78B tokens). Under this stronger training regime, the baseline achieves a perplexity of **22.15**, and Transponder achieves **19.56**.
>
> Both models are able to reach a perplexity lower than 22.50 on OWT when trained longer, yet Transponder consistently outperforms the baseline. The training step setting we are following the article of [1] and [2]. We have added these extra results and comparisons in Appendix E for clarity.
>
> **W2:** Did the authors properly tune the baseline? What is the experiment setup?
>
> **Reply:** Thank you for the insightful question. The hyperparameter settings used in our initial experiments were directly taken from the settings reported in MixLN [1] and Curse of Depth [2]. We also showed the training hyperparameters on Table 7 in the submitted manuscript. In our preliminary tests, we found that the main property that will affect the training stability is the learning rate. In our preliminary tests, we found that both Transponder and other gating-based variants remain stable even under a relatively high learning rate of 3e-3. However, the original baseline always failed in training. Therefore, for consistency, all gating-based methods (including Transponder) were trained using the same learning rate of 3e-3 in the main experiments.
>
> To directly address the reviewer’s concern, we additionally performed a comprehensive learning-rate sweep (1e-2, 3e-3, 1e-3) for three model variants:
> (1) LLaMA_widen,
> (2) SDFA [3], and
> (3) Transponder.
> Results are summarized in the table below.
>
> We observe three consistent findings across datasets and model scales:
>
> 1) LLaMA_widen (the parameter-matched baseline by widening the FFN intermediate dimension such that the parameter increase is approximately the same as the Transponder) fails to train when the learning rate is high (1e-2 or 3e-3), showing severe instability and divergence.
>
> 2) SDFA improves stability to some extent, but still collapses at very high learning rates.
>
> 3) Transponder remains stable across all tested learning rates, and in several cases even achieves better performance at higher learning rates (e.g., 60M/OWT at 1e-2 and 130M/C4 at 1e-2).
>
> These results demonstrate that Transponder is substantially more robust to large learning rates, and its performance can even benefit from a more aggressive learning-rate regime—unlike the baseline or SDFA, which both exhibit strong sensitivity and training failures under the same conditions.
>
> We have added the above text in the revised manuscript.
>
> | Model Size | Dataset     | Model       | 1e-2    | 3e-3   | 1e-3  |
> | -| -| -| -| -| -|
> | **60M**    | OWT | LLaMA_widen    | 1007.00 | 600.88  | 26.59 |
> |            |             | SDFA        | 524.93  | 22.79  | 25.18 |
> |            |             | Transponder | **22.07**   | 22.65  | 25.85 |
> | **60M**    | C4          | LLaMA_widen    | 986.52  | 540.23  | 30.05 |
> |            |             | SDFA        | 693.65  | 29.24  | 31.84 |
> |            |             | Transponder | **28.06**   | 28.53  | 32.24 |
> | **130M**   | OWT | LLaMA_widen    | 683.08  | 632.76  | 19.74 |
> |            |             | SDFA        | 610.09  | 20.44  | 18.60 |
> |            |             | Transponder | 18.49   | **17.45** | 18.26 |
> | **130M**   | C4          | LLaMA_widen    | 775.34  | 669.47  | 26.63 |
> |            |             | SDFA        | 724.76  | 302.44 | 23.92 |
> |            |             | Transponder | **21.82**   | 22.13  | 23.99 |
>
>
> [1] Pengxiang Li et al. Mix-LN: Unleashing the Power of Deeper Layers by Combining Pre-LN and Post-LN. ICLR 2025
>
> [2] Wenfang Sun et al. The Curse of Depth in Large Language Models. ICML 2025
>
> [3] Zihan Qiu et al. Gated Attention for Large Language Models: Non-linearity, Sparsity, and Attention-Sink-Free. NeurIPS 2025

---

> ### Author Response · Authors · 2025-12-03
> **Reply to Reviewer SG6Y(2/2)**
>
> **W3:** How many FLOPS are used for the baseline vs the Transponder results?
>
> **Reply:** Thanks for this question. We now provide the inference and training FLOPs and the inference latency to support our claim that Transponder adds only a small overhead. The performance below are obtained on LLaMA60M and the inference latency is evaluated on a single A100 80G GPU.
>
> **Per-token FLOPs**
>
> | Model       | Forward FLOPs/token (GFLOPs) | Training FLOPs/token (GFLOPs) |
> | ----------- | ---------------------------- | ----------------------------- |
> | Original    | 0.082                        | 0.245                         |
> | SDPA  | 0.088                        | 0.264                         |
> | Transponder | 0.084                        | 0.252                         |
>
> **Inference Latency (Throughput, seq_len = 256)**
>
> | Model        | BS=32      | BS=64      | BS=128     |
> | ------------ | ---------- | ---------- | ---------- |
> | Original     | 707.57     | 721.81     | 748.74     |
> | SDPA   | 672.98     | 688.83     | 707.66     |
> | Transponder  | 654.53     | 678.96     | 689.76     |
>
>
> Overall, the results show that Transponder adds a small compute and latency cost. The FLOPs increase is about 2–3%, and the inference throughput decreases by around 7–9% depending on batch size. This FLOPs overhead is smaller than that of the SDPA baseline, and the inference latency stays modest relative to the improvements achieved.
>
> **W4:** What is held constant ? It would also be interesting to know if this works beyond ppl and if it works on downstream task.
>
> **Reply:** All hyperparameters—optimizer (except for the learning rate since the LLaMA and LLaMA_widen baseline cannot train stably with a higher learning rate), batch size, warmup steps, training setup, and model architecture—are kept identical across all variants. Only the modulation mechanism (SDPA, Transponder, or Widen-Original) is changed.
>
> To further evaluate whether the improvements go beyond perplexity, we performed zero-shot downstream evaluation on GLUE and MMLU using accuracy. To improve generalization for all models in a fair manner, we extended the training to 60K steps (≈11.8B tokens) and increased max sequence length to 1024. The results are summarized in the table below, showing that Transponder achieves the highest average accuracy and the highest win-rate (6/10 tasks), demonstrating consistent gains beyond perplexity.
>
> | Dataset          | SDPA | Transponder | Widen-Original |
> |------------------|------------|-------------|----------------|
> | **CoLA (mcc)**   | 0.0000     | **0.0464**  | 0.0230         |
> | **COPA (acc)**   | 0.56       | **0.61**    | 0.58           |
> | **HellaSwag_norm** | 0.2989   | **0.3036**  | 0.2928         |
> | **MNLI (acc)**   | **0.3281** | 0.3264      | 0.3277         |
> | **MRPC (acc)**   | **0.6912** | 0.6838      | 0.6201         |
> | **QNLI (acc)**   | 0.4944     | **0.4946**  | 0.4937         |
> | **QQP (acc)**    | 0.3683     | 0.3685      | **0.3691**     |
> | **RTE (acc)**    | **0.5343** | 0.5307      | 0.4765         |
> | **SST-2 (acc)**  | 0.4908     | **0.5195**  | 0.4908         |
> | **MMLU (acc)**   | 0.22953    | **0.23002** | 0.22924        |
> | **Average (9 tasks)** | 0.4184 | **0.4315** | 0.4082         |
> | **Win-Rate (10 tasks)** | 3 / 10 | **6 / 10** | 1 / 10       |
>
>
>
>
> **W5:** Efficiency: How does the Transponder affect the training & inference latency and throughput ? Do these make the models slower?
>
> **Reply:** Please refer to the reply to Weakness 3.

---

### Official Review · Reviewer_kjNo · 2025-11-06

**Soundness:** 3
**Presentation:** 3
**Contribution:** 2
**Rating:** 6
**Confidence:** 4

**Summary:**

In this paper the authors produce adding input-aware modulation, otherwise known as gating, to all transformer layers and projections. This is done by using a sigmoid-activated low-rank layer that is multiplied with attention/ffw layer outputs as well as outputs of individual projections. The results show significant perplexity gains over a LLAMA baseline at different model scales, with minimal (1%) parameter overhead. The authors perform extensive ablations of their method, concluding that the rank can be relatively low, and that adding multiple such gating modules improves model quality.

**Strengths:**

* The experimental setup is sound

* The results show significant perplexity improvements and show improvement over a previous baseline (Laurel)

* The authors' view of adding modulation everywhere is unifying and interesting.

**Weaknesses:**

* There is related work with significant overlap that is not discussed. In particular, all the following papers use input-aware gating and show it improves model quality significantly. The authors should discuss them and describe the differences with their work.
https://arxiv.org/pdf/2409.19606
https://arxiv.org/pdf/2502.06785
https://arxiv.org/pdf/2505.06708

* The authors should show the hyperparameters they used in their final model (which modulators in Fig 1 are scalar vs channel based, what is the rank used in each case etc) are and how the 1% extra params is calculated. (In case it was mentioned and I missed it, it would help to make it more visible in the main body)

* The authors should train a baseline model with +1% extra params to compare the perplexities in a fair way.

* The authors could add some measurements and discussion on training time impact.

**Questions:**

* Were the learning rates of the baseline model tuned? This is especially relevant since the authors use 2 * sigmoid activation, which might artificially increase the learning rate and confound the results.

* How does the approach compare to pervious work referenced above?

* What happens if we ablate the intermediate (low-rank) activation?

---

> ### Author Response · Authors · 2025-12-03
> **Reply to Reviewer kjNo(1/4)**
>
> We sincerely thank the reviewer for the careful reading of our paper and for the thoughtful, constructive feedback. We reply the reviewer's comments and questions as follows:
>
> **W1:** There is related work with significant overlap that is not discussed. In particular, all the following papers use input-aware gating and show it improves model quality significantly. The authors should discuss them and describe the differences with their work. https://arxiv.org/pdf/2409.19606 https://arxiv.org/pdf/2502.06785 https://arxiv.org/pdf/2505.06708
>
> **Reply:**
> We thank the reviewer for pointing out these related articles. I have **already** discussed the third article in the article and compared our method with theirs. We will add the other two in the revised Related Work as follows.
>
> Hyper-Connections [1] generalize residual connections by maintaining a *stack* of hidden states and learning static or input-dependent matrices (“dynamic hyper-connections”) that mix features both across depth and width; in the dynamic variant, the coefficients (B(H), A_m(H), A_r(H)) are predicted from the current hyper-state and applied to all layers’ hidden vectors. DeepCrossAttention [2] builds on Generalized Residual Networks (GRN-v1/v2/v3) to form input-dependent linear combinations of *all previous layer outputs* and uses these stack-based GRNs to generate Q/K/V for each block, thereby implementing depth-wise cross-attention and dynamic layer re-composition. Gated Attention [3] systematically studies gating positions inside softmax attention and finds that applying a head-specific sigmoid gate to the SDPA output (or values) improves perplexity and stability for 1.7B–15B LLMs, primarily by adding non-linearity and query-dependent sparsity within the attention block.
>
> We also clarified the difference, innovations, and contributions here:
> Our **Transponder** differs from all three in several key aspects:
> 1. **Where the modulation acts.** Hyper-Connections and DeepCrossAttention operate over *stacks of layer outputs* and learn how to mix representations across depths (and widths) before feeding them into attention/FFN; Gated Attention modifies the *internal attention computation* (SDPA/value) but does not change how attention/FFN outputs are injected into the residual stream. By contrast, Transponder keeps the backbone attention and FFN unchanged and instead introduces *residual modulators* that directly control the mixing between each block’s output and the main residual stream at **layer-wise and channel-wise** granularity. This decouples “what is computed” (base Transformer sublayers) from “how strongly it is expressed” (modulators), which is not explicitly addressed in the above works.
>
> 2. **Local vs. cross-layer control and overhead.** Hyper-Connections [1] and DeepCrossAttention [2] explicitly maintain and re-weight a multi-layer stack, whose mixing patterns span many layers. Transponder uses a *local*, low-rank contextual network to modulate from the current hidden state only, without storing or re-accessing the full depth history. Consequently, our modulators add ~1% parameters while still yielding large PPL improvements in the LLaMA model family; they are closer in spirit to a per-block “controller” for the residual path than to cross-layer aggregation mechanisms. (All these methods have small overhead, but the structural role of the added parameters is different.)
>
> 3. **Bio-inspired Hierarchical Meta-learning perspective.** As shown in the revised Figure 1, conceptually, our modulators are designed as a bio-inspired hirechical meta-learning mechanism: the modulators control the expression of each component's computation, analogous to how glial cells or epigenetic mechanisms regulate when and how underlying neurons or genes express their activity. Biologically, glial cells are now recognized as active regulators of information processing rather than passive support cells: they modulate synaptic efficacy, neuronal excitability, and local circuit states in a context-dependent way, providing a flexible “control layer” on top of relatively stable synaptic connectivity [4,5,6]. Thus computational neuroscientists [6,7,8] often frame glia as the optimization layer supervising neural learning. In parallel, epigenetic mechanisms adjust the accessibility and expression level of genes without changing the underlying DNA sequence, enabling long-term, experience-dependent adaptation of gene expression profiles to environmental or developmental context [9,10]. Transponder follows this neuro-glia and epigenetic perspective by decoupling what the backbone computes from how and when these computations are expressed, via lightweight modulators.

---

> ### Author Response · Authors · 2025-12-03
> **Reply to Reviewer kjNo(2/4)**
>
> Compared to standard meta-learning in machine learning, which is usually defined over a distribution of tasks and aims to “learn how to learn” across task shifts, our use of meta-learning in Transponder is hierarchical rather than across tasks. In our setting, there is a single main task (e.g., language modeling), but the optimization is decomposed into two coupled levels: 1) The base Transformer layers learn task-specific representations (the usual optimization objective). 2) The Transponder modulators learn how to help the base layers learn, by contextually modulating their activations and gradients. The modulators are not optimized to solve the prediction task directly; instead, they are optimized for their indirect effect on the performance of the base network—i.e., to improve how the base layers are trained and used.
>
>    In this sense, Transponder implements a form of hierarchical meta-learning:
>    the lower level learns the task itself, while the higher level (the modulators) learns how to organize, regulate, and adapt the learning dynamics of the base layers, analogous to “learning how to learn” but applied within a single task rather than across multiple tasks.
>
> We have added a dedicated section in the Introduction and Appendix C summarizing these methods and explicitly contrasting their cross-layer or attention-internal gating with our residual-path modulators and their bio-inspired meta-learning role.
>
> **W2:** The authors should show the hyperparameters they used in their final model (which modulators in Figure 1 are scalar vs channel based, what is the rank used in each case etc) are and how the 1% extra params is calculated. (In case it was mentioned and I missed it, it would help to make it more visible in the main body)
>
> **Reply:** Thank you for the valuable suggestion. In the original manuscript (L296–298), we stated that the default Transponder configuration is Modulator-layer-channel-scalar, i.e., each linear projection is equipped with a channel-wise modulator and a scalar modulator. We agree that these implementation details and the parameter calculation should be made more explicit in the main text.
>
> To further clarify this and avoid any misunderstanding, we have revised the manuscript to clarify all the concerns the reviewer asked.
>
> 1. **Clarification of modulators in Transponder**
>
> In the revised manuscript, we clarified that Transponder is the for *each* linear layer in the Transformer that equipped with **two contextual modulators**:
>    - a **channel-wise modulator**, whose output dimension matches that of the linear layer being modulated, and
>    - a **scalar-wise modulator**, which outputs a single scalar that modulates the entire linear-layer output.
>
> Both modulators share the same two-layer structure (low-rank projection → non-linear activation → projection), differing only in output dimensionality. We updated Figure 1 to explicitly reflect this design. For the Modulator-path variant, we keep it as an ablation in the Appendix B, since it does not yield as much improvement as the Modulator-layer variants.
>
> 2. **Hyperparameters (rank and configuration)**
>
> As stated in L237–239, the hidden dimension (rank) of all modulators in our main experiments is set to **r=8**. The remaining hyperparameters (e.g., learning rate, batch size, sequence length) are listed in Section 4.1 and summarized in Table 9 of the Appendix. We additionally highlight in the main text which modulators are scalar-based and which are channel-based in Figure 1.
>
> 3. **1% extra parameter calculation**
>    We added a dedicated paragraph in the Appendix H with the full derivation. A brief summary is provided here:
>    - Adding the modulators to each linear layer introduces:  $(768 \times 8 \times 2 + 8 \times 1) \times 4 + (768 \times 8 + 2048 \times 8 + 8 \times 1) \times 2 + (768 \times 8 + 2048 \times 8 + 8 \times 1)= 116,792$
>    - The resulting overhead across the full model is approximately **116,792 * 12 /130,000,000 = 1.08% ～ 1%**.
>
> We updated the parameter table and added the full computation process of all the models involved in the Appendix H for clarity.
>
> **W3:** The authors should train a baseline model with +1% extra params to compare the perplexities in a fair way.
>
> **Reply:** Please see the **reply to Q2** below.

---

> ### Author Response · Authors · 2025-12-03
> **Reply to Reviewer kjNo(3/4)**
>
> **Q1:** Were the learning rates of the baseline model tuned? This is especially relevant since the authors use 2 * sigmoid activation, which might artificially increase the learning rate and confound the results.
>
> **Reply:** Thank you for the insightful question. The learning rates used in our initial experiments were directly taken from the settings reported in MixLN [11] and Curse of Depth [12]. In our preliminary tests, we found that both Transponder and other gating-based variants remain stable even under a relatively high learning rate of 3e-3. However, the original baseline always failed in training. Therefore, for consistency, all gating-based methods (including Transponder) were trained using the same learning rate of 3e-3 in the main experiments.
>
> To directly address the reviewer’s concern, we additionally performed a comprehensive learning-rate sweep (1e-2, 3e-3, 1e-3) for three model variants:
> (1) Original_widen,
> (2) SDPA [3], and
> (3) Transponder.
> Results are summarized in the table below.
>
> We observe three consistent findings across datasets and model scales:
>
> 1) LLaMA_widen (the parameter-matched baseline by widening the FFN intermediate dimension such that the parameter increase is approximately the same as the Transponder) fails to train when the learning rate is high (1e-2 or 3e-3), showing severe instability and divergence.
>
> 2) SDPA improves stability to some extent, but still collapses at very high learning rates.
>
> 3) Transponder remains stable across all tested learning rates, and in several cases even achieves better performance at higher learning rates (e.g., 60M/OWT at 1e-2 and 130M/C4 at 1e-2).
>
> These results demonstrate that Transponder is substantially more robust to large learning rates, and its performance can even benefit from a more aggressive learning-rate regime—unlike the baseline or SDPA, which both exhibit strong sensitivity and training failures under the same conditions.
>
> | Model Size | Dataset     | Model       | 1e-2    | 3e-3   | 1e-3  |
> | - | -| -| - | -| - |
> | **60M**    | OWT         | LLaMA_widen    | 1007.00 | 600.88  | 26.59 |
> |  | | SDPA        | 524.93  | 22.79  | 25.18 |
> |  |  | Transponder | **22.07**   | 22.65  | 25.85 |
> | **60M**    | C4          | LLaMA_widen    | 986.52  | 540.23  | 30.05 |
> |  |   | SDPA        | 693.65  | 29.24  | 31.84 |
> |  |             | Transponder | **28.06**   | 28.53  | 32.24 |
> | **130M**   | OWT | LLaMA_widen    | 683.08  | 632.76  | 19.74 |
> | |             | SDPA        | 610.09  | 20.44  | 18.60 |
> | |             | Transponder | 18.49   | **17.45** | 18.26 |
> | **130M**   | C4          | LLaMA_widen    | 775.34  | 669.47  | 26.63 |
> |            |             | SDPA        | 724.76  | 302.44 | 23.92 |
> |            |             | Transponder | **21.82**   | 22.13  | 23.99 |
>
> We have incorporated these results and the analysis in Appendix Table 10.
>
> **Q2:** The authors should train a baseline model with +1% extra params to compare the perplexities in a fair way.
>
> **Reply:** Thank you for this insightful suggestion. To ensure a fully fair comparison, we trained a parameter-matched baseline by widening the FFN intermediate dimension such that the parameter increase is approximately +1%, comparable to the overhead introduced by Transponder. We denote this model as LLaMA_widen.
>
> Below we report the updated results for the four variants—LLaMA (baseline), LLaMA_widen, SDPA, and Transponder—across 60M, 130M, and 250M model scales. We keep this table in the main paper and move the remaining ablations to the Appendix G to save space.
>
> | Datasets    | Model            | LLaMA60M  | LLaMA130M | LLaMA250M |
> | -| -| - | -| -|
> | Openwebtext | LLaMA (baseline) | 26.56     | 19.27     | 17.28     |
> |             | LLaMA_widen      | 26.59     | 19.74     | 17.10     |
> |             | SDPA             | 22.79     | 20.44     | 15.65     |
> |             | Transponder      | **22.07** | **17.45** | **14.93** |
> | C4          | LLaMA (baseline) | 30.31     | 26.73     | 21.92     |
> |             | LLaMA_widen      | 30.28     | 26.63     | 21.79     |
> |             | SDPA             | 29.24     | 23.92     | 19.12     |
> |             | Transponder      | **28.06** | **21.82** | **18.72** |
>
> We additionally evaluate the same four models at the 1B parameter scale on OpenWebText. These results are included in the Table 1 for completeness.
>
> | Model            | LLaMA1B |
> | -| -|
> | LLaMA (baseline) | 14.63   |
> | LLaMA_widen      | 14.55   |
> | SDPA             | 13.82   |
> | Transponder      | **12.51**|
>
> Across all model sizes (60M → 1B) and datasets (OpenWebText, C4), the widened baseline (+1% params) performs very similarly to the original baseline, confirming that simple parameter scaling alone does not yield meaningful perplexity gains. In contrast, Transponder consistently improves perplexity at all scales, demonstrating that its benefits come from its contextual modulation mechanism rather than from extra parameters.

---

> ### Author Response · Authors · 2025-12-03
> **Reply to Reviewer kjNo(4/4)**
>
> **Q3:** The authors could add some measurements and discussion on the training time impact.
>
> **Reply:** Thanks for this insightful question. We now provide the inference and training FLOPs and the inference latency to support our claim that Transponder adds only a small overhead. The performance below is obtained on LLaMA60M, and the inference latency is evaluated on a single A100 80G GPU.
>
> **Per-token FLOPs**
>
> | Model       | Forward FLOPs/token (GFLOPs) | Training FLOPs/token (GFLOPs) |
> | ----------- | ---------------------------- | ----------------------------- |
> | Original    | 0.082                        | 0.245                         |
> | SDPA | 0.088                        | 0.264                         |
> | Transponder | 0.084                        | 0.252                         |
>
> **Inference Latency (Throughput, seq_len = 256)**
>
> | Model        | BS=32      | BS=64      | BS=128     |
> | ------------ | ---------- | ---------- | ---------- |
> | Original     | 707.57     | 721.81     | 748.74     |
> | SDPA   | 672.98     | 688.83     | 707.66     |
> | Transponder  | 654.53     | 678.96     | 689.76     |
>
>
> Overall, the results show that Transponder adds a small compute and latency cost. The FLOPs increase is about 2–3%, and the inference throughput decreases by around 7–9% depending on batch size. This FLOPs overhead is smaller than that of the SDPA baseline, and the inference latency stays modest relative to the improvements achieved.
>
>
> **Reference:**
>
>
> [1] https://arxiv.org/pdf/2409.19606 "HYPER-CONNECTIONS"
>
> [2] https://arxiv.org/pdf/2502.06785 "DeepCrossAttention: Supercharging Transformer Residual Connections"
>
> [3] https://arxiv.org/pdf/2505.06708 "Gated Attention for Large Language Models: Non-linearity, Sparsity, and Attention-Sink-Free"
>
> [4] Fields, R. D. (2004). The other half of the brain. Scientific American.
>
> [5] Araque, A., Carmignoto, G., Haydon, P. G., Oliet, S. H. R., Robitaille, R., and Volterra, A. (2014). Gliotransmitters travel in time and space. Neuron.
>
> [6] De Pittà, M., and Brunel, N. (2016). Astrocytes: orchestrating synaptic plasticity? Neuroscience.
>
> [7] De Pittà et al., 2011. A Tale of Two Stories: Astrocyte Regulation of Synaptic Depression and Facilitation. PLoS Computational Biology.
>
> [8] Volman et al., 2012. Computational models of neuron–astrocyte interaction in synaptic transmission. Frontiers in Computational Neuroscience.
>
> [9] Bird, A. (2007). Perceptions of epigenetics. Nature.
>
> [10] Sweatt, J. D. (2013). The emerging field of neuroepigenetics. Neuron.
>
> [11] Pengxiang Li et al. Mix-LN: Unleashing the Power of Deeper Layers by Combining Pre-LN and Post-LN. ICLR 2025
>
> [12] Wenfang Sun et al. The Curse of Depth in Large Language Models. ICML 2025

---

### Author Response · Authors · 2025-12-03
**Summary of the Rebuttal**

We sincerely thank all reviewers for their time and constructive feedback, which is very helpful for improving our work. We are glad that the reviewers recognize:

(1) the **well-motivated problem** of static residual scaling in Transformers and the need for input-aware modulation (Reviewers SG6Y, Kpr8, SMQN);

(2) the **conceptual contribution** of Transponder in decoupling representation from control via input-dependent modulators and providing a unifying “modulate everywhere” view of residual pathway control (Reviewers kjNo, SG6Y, 7pnL);

(3) the **simple and modular design** that can be plugged into standard Transformers with minimal changes and less than 1% extra parameters and modest overhead (Reviewers kjNo, Kpr8, SMQN);

(4) the **clear and well-organized writing**, which makes the method and its integration easy to follow (Reviewer SG6Y);

(5) the **strong empirical results and analysis**, including 5–15% perplexity reductions across LLaMA scales, improvements over the Laurel baseline, stabilized training in challenging Post-LN setups, and extensive ablations and visualizations on placement, granularity, hidden size, and modulation patterns (Reviewers kjNo, SG6Y, 7pnL, Kpr8, SMQN).

We can see from the reviewers’ weaknesses and questions that most concerns focus on hyperparameter choices, additional baselines, differences from existing methods, and training/inference efficiency. We have addressed these points thoroughly in our rebuttal.

In particular, we:

* Added the extra related work suggested by the reviewers. (Reviewer **kjNo** W1)
* Clarified the differences between Transponder and gating-based methods. (Reviewer **kjNo** W1, Reviewer **7pnL** W1, Reviewer **SMQN** W1)
* Clarified the hyperparameter settings and how the additional parameters are computed. (Reviewer **kjNo** W2, Reviewer **SG6Y** W2)
* Added a learning-rate sensitivity study. (Reviewer **kjNo** Q1, Reviewer **SG6Y** W2)
* Introduced a new baseline **LLaMA_widen** with a parameter count matched to Transponder. (Reviewer **kjNo** Q2, Reviewer **Kpr8** W4, Reviewer **SMQN** W4)
* Added analyses of training and inference FLOPs and inference latency. (Reviewer **kjNo** Q3, Reviewer **SG6Y** W3 & W5, Reviewer **Kpr8** W2)
* Included experiments on downstream tasks. (Reviewer **SG6Y** W4, Reviewer **Kpr8** W1)
* Clarified why performance degrades or training fails in some configurations. (Reviewer **7pnL** W2)
* Explained why Transponder leads to more stable training dynamics. (Reviewer **7pnL** Q1, Reviewer **Kpr8** W3, Reviewer **SMQN** W3)
* Added results on larger models. (Reviewer **Kpr8** W1)
* Performed an ablation study on the up and gate projections. (Reviewer **Kpr8** W5)
* Extended the study to other Transformer variants. (Reviewer **SMQN** W2)
* Corrected minor typographical and grammatical issues. (Reviewer **SMQN**)

In conclusion, we believe that the additional experiments, analyses, and clarifications provided in the rebuttal adequately address the main concerns raised by the Reviewers. We again sincerely thank all reviewers for their thoughtful and constructive feedback, and we are also grateful to the Area Chair for their careful handling and evaluation of our submission.

---

### Meta-Review · Area_Chair_xLXG · 2026-01-20

**Summary:**

This paper receives scores of 6, 6, 4, 4, 2. The main concerns are about runtime, experiment setup, and theoretical analysis. After reading the author's response, some critical concerns remain. (1) It is unclear whether, using adequate training iterations and suitable learning rate, the proposed method is still significant (e.g., compared with SDPA). (2) Missing evaluation on more challenging and common QA benchmarks (e.g., GPQA, MMLU, and GSM8K), which is critical to the practicality. I suggest the author use these common benchmarks in all experiments beyond ppl. (3) The conceptual novelty is incremental against existing gating-based method.

**Reviewer Concerns:**

Most concerns are addressed, but some critical concerns about experiments remain.

Reviewer kjNo (Rating: 6)

(1) Discussion with related gating-based works, including hyper-connections, deep cross attention, and gated attention.

Have provided a detailed discussion. But, it is better to provide experiments comparisons on larger models and more challenging LLM benchmarks (e.g., GPQA, MMLU, and IFEval), while gated attention show significance in these common settings.

(2) Report detailed hyperparameters

Addressed.

(3) Experiment with baseline +1% extra params

Addressed.

(4) Reporting training time

Addressed.

(5) Validation on different learning rates

Addressed

(6) Ablate the intermediate (low-rank) activation.

No response.

Reviewer SG6Y (Rating: 6)

(1) It should be possible to train a llama baseline like the one in the paper to achieve less than 22.50 ppl on OWT.

The authors report results with more training data to achieve 22.50 ppl for baseline. However, according to the results, the improvement has narrowed. It is unclear whether, using enough training data, the proposed method still outperforms SDPA in terms of ppl.

(2) Did the authors properly tune the baseline ? What is the experiment setup?

Addressed by experiments on different learning-rates.

(3) How many FLOPS are used for the baseline vs the Transponder results?

Addressed

(4) What is held constant? It would also be interesting to know if this works beyond ppl and if it works on downstream taks.

Provided results on GLUE and MMLU. The results on MMLU is almost the same, while GLUE is not commonly used by recent LLMs. Evaluation on more common and challenging benchmarks is necessary, such as GSM8k, HumanEval, and Hellaswag.

(5) Efficiency

Addressed.

Reviewer 7pnL (Rating: 4)

(1) Similarity to Gating Methods.

Addressed

(2) In Table 11, the first-row results (“Modulator-path-scalar”) show abnormally high PPL (e.g., 1088 for 250M), suggesting instability.

Addressed. It may caused by aggressive the learning rate.

(3) Table 1 lacks direct comparison with the LLaMA baseline for those variants, making it hard to gauge how much each modulator improves over standard training.

Addressed.

(4) I would like to ask about the significant reduction in PPL for Table 4. What is the reason behind this performance improvement? The method mentioned in the paper still follows a Transformer-like architecture, so theoretically, with the same number of parameters, there shouldn't be such a substantial performance change.

The author provided an exploration. However, according to the exploration and experiments, this method seems to be a technique for stabilizing training but it is unclear whether it can improve the results under adequate training iterations and suitable hyperparameters.

Reviewer Kpr8 (Rating: 4)

(1) Evaluation on larger model sizes and real downstream benchmarks. Evaluation on larger model sizes are missing.

Provided results on GLUE and MMLU. The results on MMLU is almost the same, while GLUE is not commonly used by recent LLMs. Evaluation on more common and challenging benchmarks is necessary, such as GSM8k, HumanEval, and Hellaswag.

(2) Reporting FLOPs, latency, or memory statistics.

Addressed

(3) Analysis of how contextual modulation stabilizes training or enhances representational capacity.

Addressed

(4) Comparisons against equal-parameter alternatives.

Addressed

(5) Partial overlap with LLaMA’s built-in gate projections: A single ablation (“w/o up and gate”) hints at redundancy but lacks a full quantitative study.

Addressed.

Reviewer SMQN (rating: 2)

(1) The conceptual novelty is incremental.

The author provided a discussion, but the novelty is still incremental.

(2) It is only evaluated on LLaMA backbones, without exploring its effectiveness on other Transformer variants.

Addressed by providing results on encoder–decoder basic transformer.

(3) It lacks theoretical analysis to explain why contextual modulation stabilizes training or improves performance.

Addressed.

(4) They do not compare this parameter allocation with alternative ways to use the same parameter budget (e.g., widening FFN or adding lightweight adapters). Therefore, it is unclear whether the observed improvement is specific to the proposed mechanism or simply a result of having more capacity.

Addressed.

**Reviewer Scores:**

Reviewer SMQN may raise the rating since some of concerns are addressed.

---

### Decision · Program_Chairs · 2026-01-26

Reject